# REPRESENTATIONS IN A DEEP END-TO-END DRIVING MODEL PREDICT HUMAN BRAIN ACTIVITY IN AN ACTIVE DRIVING TASK

## ABSTRACT

Understanding how cognition and learned representations give rise to intelligent behavior is a fundamental goal in both machine learning and neuroscience. However, in both domains, the most well-understood behaviors are passive and open-loop, such as image recognition or speech processing. In this work, we compare human brain activity measured via functional magnetic resonance imaging with deep neural network (DNN) activations for an active taxi-driving task in a naturalistic simulated environment. To do so, we used DNN activations to build voxelwise encoding models for brain activity. Results show that encoding models for DNN activations explain significant amounts of variance in brain activity across many regions of the brain. Furthermore, each functional module in the DNN explains brain activity in a distinct network of functional regions in the brain. The functions of each DNN module correspond well to the known functional properties of its corresponding brain regions, suggesting that both the DNN and the human brain may partition the task in a similar manner. These results represent a first step towards understanding how humans and current deep learning methods agree or differ in active closed-loop tasks such as driving.

## 1 INTRODUCTION

Fully autonomous vehicles now share the roads with human drivers in several major cities. In contrast with most previous commercially successful applications of robotics, such as robotized warehouses (Azadeh et al., 2019), drone medication delivery in rural areas (Demuyakor, 2020), and infrastructure inspection (Lattanzi & Miller, 2017), autonomous driving systems perform the same tasks as humans do in diverse and complex environments involving dynamic interactions with other agents. Ensuring that the algorithms used in autonomous driving are effective and safe to very low margins of error poses a significant challenge. Furthermore, most autonomous driving systems use deep neural networks (DNNs). DNNs are notoriously difficult to analyze and interpret (Zhang et al., 2021), which makes verification of these desirable properties an additional problem. However, there are also exciting opportunities: for the first time, human and artificial intelligences (AIs) are performing the same complex sensorimotor and social tasks in a shared environment. This raises many interesting questions at the intersection of artificial intelligence and human neuroscience: to what extent do the human brain and DNNs rely on similar representations when integrating diverse sensory information to produce task-appropriate behavior? Do the human brain and DNN share any functional organization for solving complex, closed-loop tasks? Do they make similar kinds of inferences about each other when engaged in a shared task environment?

In this work, we provide some initial insights into these questions by comparing two sets of features: brain activity from human drivers recorded with functional magnetic resonance imaging (fMRI), and activations from the Learning from All Vehicles (LAV) driving DNN (Chen & Krähenbühl, 2022) when given similar inputs to those seen by the human drivers. To perform this comparison, we regress the LAV activations onto the brain activity to build a voxelwise encoding model (Naselaris et al., 2011; Nunez-Elizalde et al., 2019), then test this model's predictive power on held-out brain data. We find that the LAV voxelwise encoding model explains a substantial amount of variance in human brain activity. Furthermore, features from the modules in the LAV DNN explain variance in distinct functional regions in the brain. The known functional properties of these regions align well

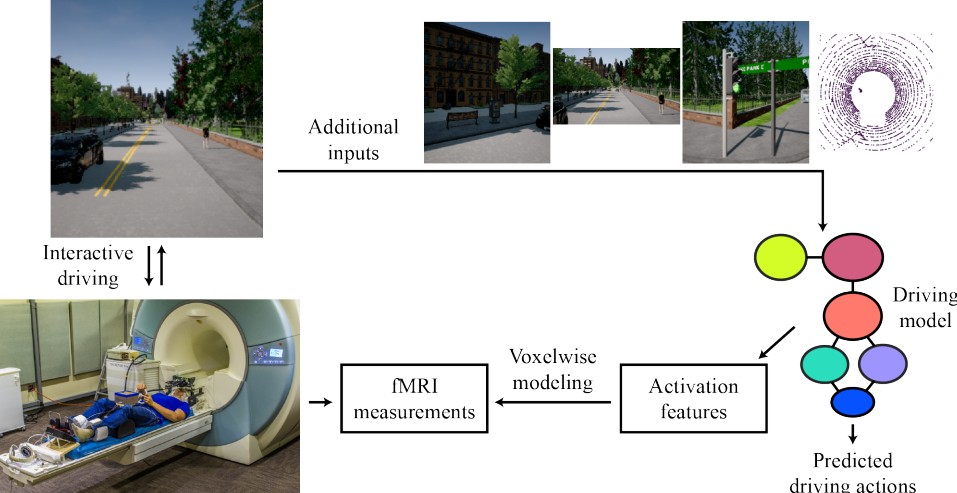

Figure 1: **Using deep neural network activations to model brain activity during a naturalistic simulated driving task.** Human subjects drove through a large virtual city in the CARLA simulator and performed a taxi-driver task while brain activity was recorded with functional magnetic resonance imaging (fMRI). The recorded experiment state was used to generate the image and LiDAR sensor inputs that were then passed to Learning from All Vehicles (LAV), a deep neural network driving model. Activations in the LAV model in response to these inputs were regressed onto brain activity to produce a predictive encoding model of brain activity during the driving task. This model can be used to investigate and quantify the alignment of the neural network driving model to the representations the human brain employs during driving.

with the corresponding modules. Our results are an exciting first step towards investigating the cognitive and representational basis for human and AI driving by leveraging each to better understand the other.

## 2 RELATED WORK

### 2.1 HUMAN BRAIN ACTIVITY DURING DRIVING

The cognitive processes underlying human driving behavior have been studied in many experiments and with a variety of brain imaging modalities, including fMRI, electroencephalography (EEG), functional near-infrared spectroscopy (fNIRS), and magnetoencephalography (MEG) (Haghani et al., 2021). Of these modalities, fMRI provides the highest spatial resolution and enables the localization of brain activity (Gross, 2019), and has been used to localize various aspects of driving to functional brain regions. Both Schweizer et al. (2013) and Spiers & Maguire (2007) have used fMRI to examine brain activity from subjects performing simulated driving tasks. Navarro et al. (2018) pooled these and seven other fMRI studies of driving to find characteristic patterns of brain activity for low-level tasks such as turning, medium-level tasks such as planning to overtake, and high-level tasks such as route planning, and related these patterns to conceptual (not quantitative or predictive) models of human driving. Unfortunately, these studies only used contrast-based fMRI analysis methods to localize driving-related activity to certain brain regions, and did not investigate hypotheses about specific representations and computations in the brain during driving. Also, these studies provide no insights into AI driving algorithms.

### 2.2 DNNs AS ENCODING MODELS OF BRAIN ACTIVITY

Encoding models are a powerful method for the analysis of brain activity. In this approach, the time series of brain activity is modelled as a function of stimulus and task feature time series. Encoding models have been extensively used to study how the brain represents visual inputs. Features used to fit encoding models of brain activity in these studies range from simple transformations of the visual

input, such as Gabor wavelets (Carandini et al., 2005), to deep neural networks trained on vision tasks (Agrawal et al., 2014; Güçlü & Van Gerven, 2015; Takagi & Nishimoto, 2023). Because previous studies using DNN-based encoding models have focused on visual processing, they used data from tasks in which subjects observed stimuli passively. Here we aim to apply the encoding model approach to data from an active sensorimotor task in a naturalistic environment. While vision is an important part of this task, active driving also involves planning, motor control, and social interaction with other agents.

## 3 VOXELWISE MODELING WITH DEEP NEURAL NETWORKS FOR DRIVING

To compare deep neural network activations with brain activity, we first collected a dataset of driving behavior and brain activity from human subjects. We then used the recorded behavior from this dataset to generate a time series of appropriate inputs to the LAV driving network. Next, we gave these inputs to the LAV driving network, and extracted features representing the activity of the LAV network. Finally, we built voxelwise models of brain activity by regressing the LAV driving network features against the brain activity.

### 3.1 MEASURING BRAIN ACTIVITY WITH fMRI

We used functional magnetic resonance imaging (fMRI) to record brain activity from three subjects performing a taxi-driver task in a simulator. We used Unreal Engine 4 and the CARLA plugin to implement a driving simulator that contains a large $2 \times 3$ km virtual city populated by dynamic pedestrians and vehicles. Prior to recording, subjects learned the layout of the world. During recording, subjects controlled a virtual car with an MR-compatible steering wheel and pedals (illustrated in Fig.1), performing the taxi-driver task, in which they were cued to navigate to particular locations. Blood-oxygenation-level dependent (BOLD) (Ogawa et al., 1990) activity from the brain was recorded by the MRI scanner as subjects performed the task. Data were collected in 11-minute runs (180 minutes for subjects 1 and 2, and 110 minutes for subject 3) at a temporal resolution of 2.0045 seconds and a voxel (3D pixel) size of $2.24 \times 2.24 \times 3.5$ mm$^3$ (matrix size = $100 \times 100$ voxels, 30 axial slices). Please refer to appendix A.3 for more details on data collection and preprocessing. The experimental procedures were approved by the Institutional Review Board at [redacted], and written informed consent was obtained from all subjects.

### 3.2 MEASURING DEEP NEURAL NETWORK ACTIVITY

#### 3.2.1 DNN DRIVING MODEL

For this work, we chose to examine a pre-trained driving deep neural network developed by Chen & Krähenbühl (2022), Learning from All Vehicles (LAV). LAV was developed for the sensors track of the CARLA Leaderboard challenge (version 1.0), where teams compete to submit algorithms that can complete a sequence of driving scenarios while obeying traffic rules and avoiding collisions. LAV achieved first place in the 2021 edition of the challenge, with a route completion rate of 94% (Chen & Krähenbühl, 2022). It uses an imitation learning-based training objective to mimic "expert" driving behavior provided by the built-in CARLA AI driver. The CARLA AI driver uses ground-truth simulator state to generate appropriate driving actions, and LAV attempts to imitate its behavior with only RGB images and LiDAR as inputs. LAV was trained on CARLA version 0.9.10, which is slightly different than the modified version of CARLA we used in our human subject experiments, but we verified that LAV performed reasonable inferences when run with inputs from our simulator (see Fig.2**a** for example outputs on our data).

The LAV DNN consists of an end-to-end architecture, but is split into different distinct modules that each perform a specific sub-task for driving (encouraged via extra module-specific objective functions during training). We used this architecture to conceptually group the measured DNN activations by the sub-task with which they are associated. In addition to processing the RGB and LiDAR inputs and producing control outputs, these modules also include "intermediate" representations such as a bird's-eye-view of the local environment. The modules are as follows:

1. **Semantic segmentation:** This module predicts the semantic class of visual features in RGB images from the vehicle's cameras. It receives three images as input, one from a

camera facing forwards and two at yaw offsets of $\pm 60$ degrees. It categorizes each pixel in these images as one of 5 semantic classes: roads, road markings, vehicles, pedestrians, and other. The architecture is an ERFNet (Romera et al., 2017), a type of convolutional neural network specialized for segmentation.

2. **Bird's-eye-view (BEV) perception:** This module predicts local scene features, including the layout of the road and the locations of other vehicles, in a BEV reference frame fixed on the ego vehicle. It receives 3D LiDAR points as input, where points that lie within the field of view of the RGB cameras are "painted" with their semantic class label from the semantic segmentation module. The architecture is split into two learned components: a point pillar net and a perception backbone. The point pillar splits painted LiDAR points into a 320 x 320 grid of the environment surrounding the ego vehicle as seen from above, and returns a feature vector for each bin of the grid. (Since the point pillar network operates on an unordered and variable-length set of LiDAR points, we only include the final grid-based feature vector in the features extracted from the point pillar.) The perception backbone then passes these grid features through a series of convolutional layers to produce a feature map, which is decoded into a semantic segmentation of the local road layout as well as the position and orientation of other vehicles, all of which are in a top-down, bird's-eye-view perspective.

3. **Planning:** This module generates a trajectory plan for the ego vehicle. It takes as input the BEV feature map from the previous module, cropped and centered on the ego vehicle. This feature map is first passed through a ResNet-18 (He et al., 2016), which performs driving plan-agnostic processing. Then, for each of 6 possible high-level commands (turn left / right, go straight through an intersection, stay in the current lane, and lane change left / right), a gated recurrent unit (GRU) network predicts the future position in 0.25 s intervals (up to a planning horizon of 2.5 s). Finally, a second GRU network is used to refine the intermediate set of trajectories into a final plan conditioned on a target waypoint indicating the desired route. Since the human drivers in our experiment are not following a fixed route, we generate target waypoints for the planning module using the position of the human driver 40 m further along their future trajectory.

4. **Trajectory prediction:** This module generates predicted trajectories for each nearby detected vehicle. The inputs and architecture are exactly the same as in the planning module, except that the BEV feature map is cropped and centered around the detected vehicle, and there is no target waypoint (which is unknown for the other vehicles) and therefore no final trajectory-refinement GRU. Since the true high-level command is also unknown for the detected vehicles, this module also uses a single linear layer to predict the probability of each possible command, resulting in a probability distribution over the trajectories associated with each command.

5. **Hazard detection:** This module predicts whether there is a hazard present that necessitates braking. It receives four RGB images as input (the same inputs as the semantic segmentation module, plus an additional front-facing image at a higher zoom level). The architecture is a ResNet-18 followed by a linear layer for classification. During training, the output of the ResNet-18 network is additionally passed into another network which outputs a semantic segmentation, but this is not part of inference and not included in our extracted features.

6. **Control:** This module takes the outputs of the planning, prediction, and hazard detection modules, and determines the final control action that should be applied. Specifically, if the hazard detection model predicts that braking needs to occur or if the planned trajectory for the ego vehicle intersects the predicted trajectory of another vehicle, the control module will brake to stop the car. If the control module decides that no braking is needed, it uses PID control to calculate an acceleration and steering command to follow the planned trajectory provided by the planning module. Unlike the other modules, the controller is not a DNN and has no learned parameters.

### 3.2.2 MEASURING DNN ACTIVITY

To measure the activity of the network, we used recorded simulator state data from the human subjects' driving sessions to generate RGB images and 3D LiDAR scans at 15 frames per second (FPS). There are four cameras, two pointing forwards (one with a smaller field of view and longer

focal length) and two pointing at an offset angle of 60 degrees to the left and right of the forward cameras. Because LAV does not have any form of memory other than selecting the current target waypoint from a pre-defined route, we used the recorded human data as the input at every frame, and ignored LAV's final control outputs without disrupting the operation of the LAV network. Doing so enabled us to measure the activation of the LAV network in response to the same simulator state encountered by the subjects and align the activity of the LAV DNN and the human brain activity for our models.

Across all modules, the LAV DNN contains on the order of $10^8$ measurable values that vary with the input data. This large number makes it computationally intractable to store the activations from all units and regress them against brain activity. To overcome this computational challenge and reduce the dimensionality of the LAV network activations, for each module except the controller, we used a sparse random projection (SRP) of $k$ components per module (Li et al., 2006), using the implementation in Scikit-learn to generate the random projection matrix (Pedregosa et al., 2011). SRPs approximately preserve the distance between points with high probability and dramatically speed up linear regression (Woodruff et al., 2014). We provide a proof in appendix A.1 that solving ridge regression with features projected with an SRP results in an approximate solution to the original ridge regression problem with high probability, where choosing $k$ large enough results in low approximation error and high success probability. We used the same $k$ for each module because it controls for the number of regression features when comparing encoding model performances across modules. In this work, we found $k = 20,000$ to produce stable regression results across multiple random projections.

For the controller module, we used the planned trajectory and brake probability for a total of 121 features, and therefore did not apply any dimensionality reduction. The planned trajectory is in a BEV reference frame.

## 3.3 VOXELWISE MODELING

We applied the voxelwise modeling (VM; Naselaris et al. (2011); Huth et al. (2012; 2016)) framework to the features extracted from LAV to determine whether a DNN and the human brain rely on similar representations while driving, and whether the DNN and the human brain share any functional organization for active driving. In VM, the time series of activity in each brain voxel is modeled as a linear combination of the time series of all features. Models are evaluated by predicting brain activity on a held-out dataset. High prediction performance by a set of features, or feature space, suggests that the brain represents information as parameterized by that feature space (Naselaris et al., 2011). VM has enabled many new insights about brain activity, such as in experiments where subjects listen to narrative stories (Huth et al., 2016; Deniz et al., 2019) or watch movies (Nishimoto et al., 2011; Huth et al., 2012).

In this work, we treat each module of the LAV network as a separate feature space. To capture the contributions of each module to the final predictive performance of the encoding model, we used banded ridge regression (Nunez-Elizalde et al., 2019; Dupré la Tour et al., 2022), which imposes a different ridge regularization parameter on each feature space. After regression, the overall model performance was quantified by computing the $R^2$ (explained variance) between predicted and actual activity in each voxel on a held-out test set, and individual model performances for each of the feature spaces were determined by partitioning the $R^2$ between them (Pratt, 1987; Dupré la Tour et al., 2022). Intuitively, each of the feature spaces can be considered as a hypothesis of how information may be represented in (some part of) the brain; the split $R^2$ for each feature space provides a test for its corresponding hypothesis.

The BOLD signal captured by fMRI does not respond to stimuli instantaneously but rather over a period of several seconds (the hemodynamic response function). We modeled this by implementing a finite impulse response (FIR) filter with four delays of 2 s for each LAV feature, capturing up to 8 s in the past. The best-performing feature weights, regularization parameters, and FIR filter shape were empirically selected by cross-validating over 10,000 random samples.

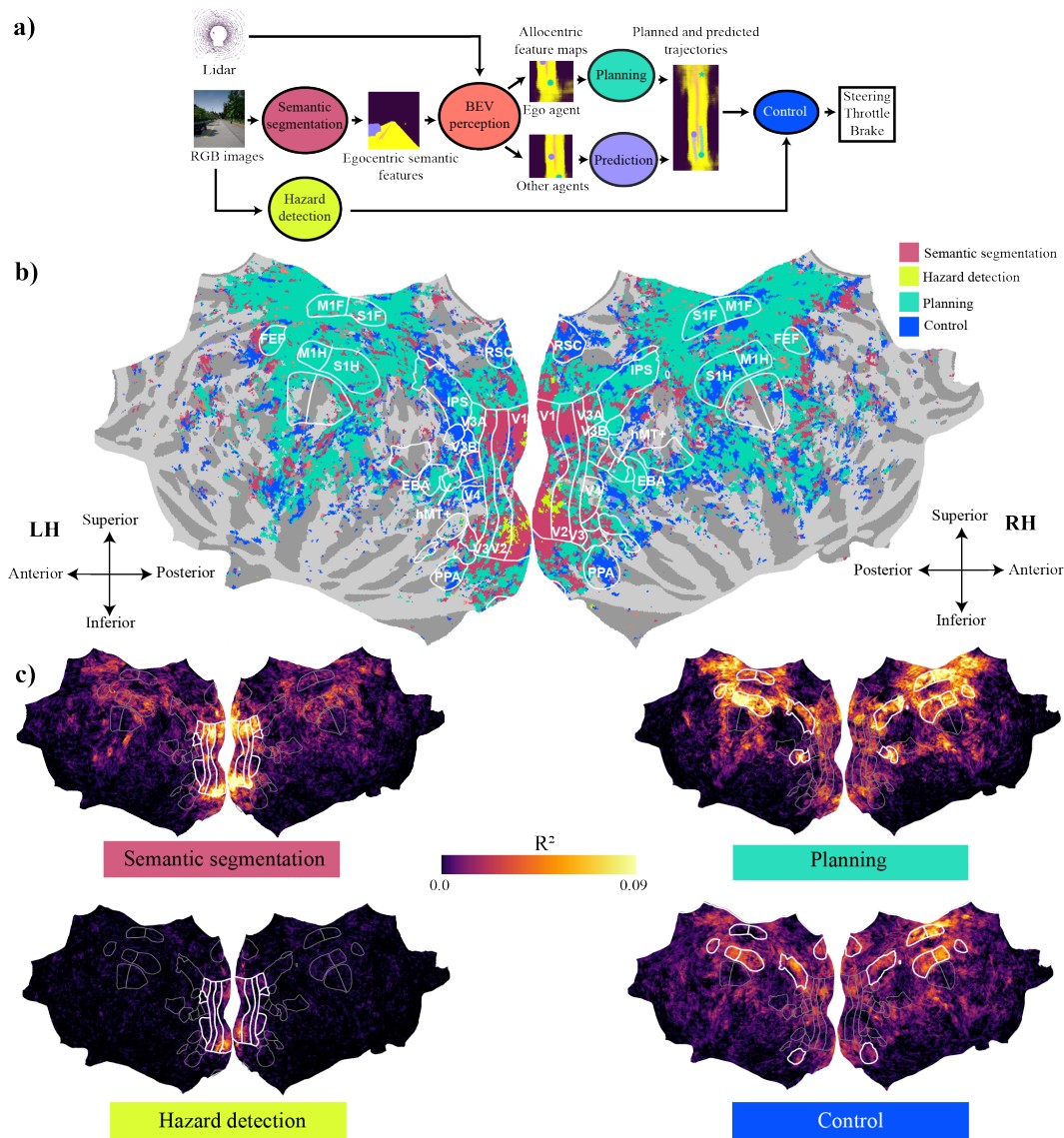

Figure 2: **Features from different modules in a deep neural network driving model predict activity in distinct functional brain networks.** a): the architecture of the Learning from All Vehicles (LAV) neural network. Each colored oval represents a different module, with their inputs and outputs shown with rectangles. Activations from each module in response to the stimuli seen by human subjects were used to fit voxelwise encoding models to the recorded brain activity. Model performance was quantified as the explained variance ($R^2$) in brain activity on a held-out dataset not used during model fitting. b): A group-level map of the best performing LAV modules across the flattened cortical surface. Voxels with a p-value of $< 0.01$ in at least 2 subjects are shown in the color corresponding to the module with the highest explained variance. Features from the semantic segmentation and hazard detection networks, which process RGB images, best explain variance in visual areas V1-V4. The planning module best explains variance in the sensorimotor areas and the intraparietal sulcus (IPS). The control module has a similar pattern of explained variance to the planning module but additionally provides a good fit to the retrosplenial cortex (RSC) and parahippocampal place area (PPA). The bird's-eye-view (BEV) perception module and the planning module for other vehicles do not explain variance in consistently localized regions across subjects. c): Group-level encoding model performance for the semantic segmentation, hazard detection, planning, and control modules across all subjects. The consistent patterns of brain activity explained by each of these modules suggests that humans and the driving network share an analogous organization of cognitive information in the active driving task.

## 4 RESULTS AND DISCUSSION

Our encoding model explains variance across much of the brain, including much of the visual and motor cortices, as illustrated in Fig.2. The model also explains brain activity in parts of the prefrontal cortex. In contrast, passive vision-only experiments only engage visual perceptual areas, and encoding models for these experiments are limited to explaining variance in these areas (Yamins et al., 2014; la Tour et al., 2021; Takagi & Nishimoto, 2023).

To determine whether the driving DNN shares any functional organizational principles with the brain, we used the split $R^2$ scores to identify the best-performing driving DNN module for every voxel (Fig.2**b**). Quantitatively, we find that the spatial organization of the best-performing modules for each voxel is non-random in all three subjects (see appendix A.3 for more details on statistical tests). Qualitatively, We find that earlier LAV modules, such as the semantic segmentation module, best explain brain activity in visual areas, whereas later modules, such as the planning module, best explain brain activity in sensorimotor and higher-level cognitive brain regions. Overall, this result suggests that the representations learned by the driving DNN may be similar to those used by the human brain, despite significant differences in perceptual inputs (human drivers have no LiDAR-like active sensing perception modalities, and our subjects were able to see only the image from a single camera with a smaller field of view than the four cameras used by LAV), training data (humans learn to drive primarily through closed-loop control rather than imitation learning), and physical substrate (real neurons behave very differently from artificial neural network units) between the two systems. In the following sections, we describe the predictive performance of each driving network module in more detail.

### 4.1 SEMANTIC SEGMENTATION AND HAZARD DETECTION

The semantic segmentation and hazard detection modules, which both process RGB images from the vehicle cameras, explain brain activity almost exclusively in the posterior visual cortex. This pattern of explained variance is qualitatively similar to previous encoding model studies that used features derived from convolutional neural networks (CNNs) trained on vision tasks such as image recognition (la Tour et al., 2021; Güçlü & Van Gerven, 2015). The hazard detection network explains more variance in the inferior part of the visual cortex, which corresponds to the upper half of the visual field. This bias may be due to the fact that, when approaching a vehicle, participants are likely to shift their gaze towards the ground. Because this downward shift moves the stimulus image upwards in the visual field, it is likely correlated with a consistent signal in the upper visual field. Another possible explanation is that traffic lights are located in the upper right part of the screen, and the hazard detection module is responsible for detecting red traffic lights that would require braking. Together, these results suggest that the visual components of a DNN in an active driving task learn similar representations to human visual processing during driving.

### 4.2 BEV PERCEPTION

The BEV perception module explains brain activity in punctate locations around the sensory and motor cortex and the anterior IPS (Fig.3**a**). (Because of the punctate nature of these regions and anatomical variability across subjects, they do not project to the same locations on the group-level map in Fig.2**b**). The performance of this module in explaining somatosensory activity is likely due to correlations between the layout of the local environment (e.g. an upcoming turn) and the driver's control actions. Because the IPS is implicated in coordinate transformations (Grefkes et al., 2004), and the BEV perception module operates in a top-down reference frame instead of an egocentric one, it is likely that these two processes are correlated, enabling the BEV perception module to predict IPS activity.

### 4.3 PLANNING

The planning module best explains activity in voxels in the somatosensory and motor cortices, the intraparietal sulcus (IPS), and anterior visual areas such as the extrastriate body area (EBA). Well-explained regions of the somatosensory and motor cortices include the primary motor (M1H/F) and primary sensory (S1H/F) cortex for hand and foot, as well as supplementary hand and foot motor areas and the supplementary eye fields. These regions are directly responsible for planning

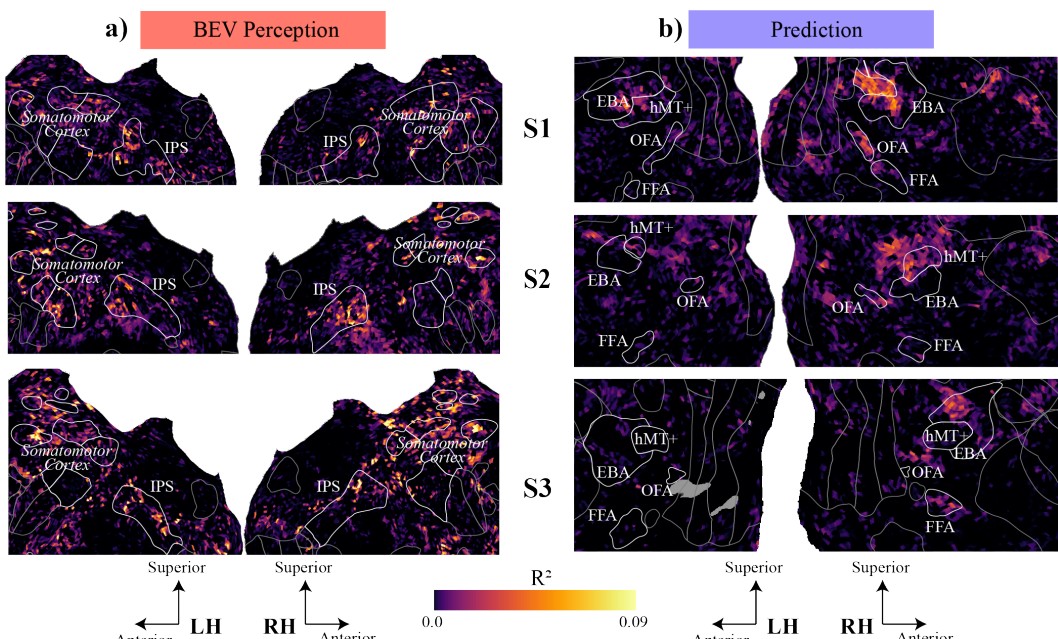

Figure 3: **Features from modules implicated in social navigation explain variance in known functional regions in the brain that perform similar tasks.** In the LAV architecture, the bird's-eye-view (BEV) perception module (left) predicts the layout of the road and other vehicles around the driver from a top-down viewpoint, while the prediction module (right) predicts the trajectories of other nearby vehicles. Unlike the other LAV modules, these are not directly connected to either sensory inputs or control outputs of the human driver, and their features are thus more abstract sets of driving-relevant information relevant to deciding how to navigate around other vehicles. a): In all three subjects, the BEV perception features explain variance in anterior intraparietal sulcus (IPS), which is implicated in coordinate transformations. They also explain variance in punctate somatomotor cortex locations. b): the prediction module explains variance in human middle temporal cortex (hMT+)/extrastriate body area (EBA) complex, and in the occipital face area (OFA) and fusiform face area (FFA). The FFA and OFA are canonically implicated in the perception of faces, while the EBA/hMT complex is canonically implicated in the perception of body parts and biological motion. These results suggest that the LAV modules are specialized in a similar manner to the functional brain regions mediating driving.

and producing motor actions to physically control the simulated vehicle. The IPS is involved in perceptual-motor coordination (Grefkes et al., 2004; Grefkes & Fink, 2005). In addition to these known functional regions, the planning module also explains some activity in the lateral prefrontal cortex.

The planning module processes local features around the subject's vehicle to output a planned trajectory based on these features. Since the driver produces control actions to follow their intended trajectory, it is intuitive that the planning and control modules provide the best fit to motor and supplementary motor areas — no LAV module contains explicit information about steering and acceleration outputs, and the features from the other modules are not as well-correlated with motor control as the planned trajectory. The IPS is known to be responsible for coordinate transformations (Grefkes et al., 2004); since like the BEV perception module the planning module operates in a bird's-eye-view reference frame, it may make use of representations similar to those found in the IPS. However, we note that overall activity in the IPS is better explained by the planning module than the BEV perception module. It is likely that the IPS simultaneously supports multiple processes for active driving, and features in our model related to coordinate frame transforms and perceptual-motor integration may both be associated with activity in the IPS. Finally, the explained variance in high-level visual areas is likely due to the correlations between certain semantic visual features (e.g.

the presence of an intersection or other vehicles) and the local BEV features and appropriate driving plan.

### 4.4 TRAJECTORY PREDICTION

The trajectory prediction module explains variance in multiple anterior visual regions, including the human middle temporal cortex (hMT+), extrastriate body area (EBA), occipital face area (OFA), and fusiform face area (FFA) (Fig.3**b**). (Similar to the well-predicted regions by the BEV perception module, some of these regions, particularly EBA, do not project well to a group-level space). The EBA is canonically implicated in the perception of body parts (Downing et al., 2001) and biological motion (Astafiev et al., 2004), the FFA in the perception of faces (Kanwisher & Yovel, 2006), and hMT+ in the perception of optic flow (Morrone et al., 2000). Note that while the FFA is canonically a face perception area, some evidence suggests that it may also function as a general expertise area for object perception (Gauthier et al., 2000; Tarr & Gauthier, 2000). Because vehicles are an object category of particular importance during driving, the FFA (and associated regions for the perception of other agents) may be recruited in this capacity while driving (Ross et al., 2018).

Because this module is downstream of other modules that receive visual inputs, explained activity in high-level vision areas may indicate alignment between the intermediate representations of other agents in the LAV architecture and the representations of human drivers. However, even in its best-performing regions, the trajectory prediction module explains less variance than the planning module, even though the planning module does not explicitly represent information about other drivers. This suggests that the representations used by the trajectory planning module may not be well-aligned with those used by the human brain. Reasoning about the behavior of other agents is a problem that has proven especially challenging in autonomous driving (Peters et al., 2024). Our results suggest that the strategy of the LAV algorithm, which projects all agents into a common representation and predicts trajectories for the ego vehicle and other vehicles in the same way, may not be a good match for how humans represent and reason about other drivers.

### 4.5 CONTROL

The control module best explains activity in a similar network of brain regions as the planning module, namely the somatosensory and motor cortices, the IPS, and some anterior visual regions. This similarity is likely because the trajectory plan that the control module attempts to follow is produced by the planning module and is therefore highly correlated with some of the features in the planning module. In the anterior visual cortex, the controller module slightly outperforms the planning module in the retrosplenial cortex (RSC) and the parahippocampal place area (PPA), two visual scene perception regions. This is likely because while the control and planning module features are both correlated with the contents of the local scene, the trajectory plan and braking probability features from the control module are more succinct than the high-dimensional representation in the planning module, and might therefore provide a stronger correlation with certain aspects of visual scene perception encoded in the RSC and PPA.

### 4.6 LIMITATIONS AND FUTURE WORK

In our experimental paradigm, the human subject controls the vehicle and therefore the stimulus (rendered RGB images from the driving simulator). This interactivity produces a naturalistic stimulus distribution, whereas previous work on comparing visual representations between the brain and CNNs (Agrawal et al., 2014; Yamins et al., 2014) have all used tightly controlled stimuli. While having an interactive task allowed us to gain insights about an active sensorimotor task, it also results in strong correlations between task state, brain activity, and driving DNN activations. These strong correlations allow for brain activity to be predicted from driving DNN activations even if algorithms and representations are not aligned between the DNN and the brain. For example, if the subject always stops their car when another vehicle is directly in front, then activity in the feet motor areas (and therefore the accelerator/brake pedals) will be predictable from a representation that encodes the presence of another vehicle even though the feet motor areas do not actually encode this information. An example of this effect is shown in Fig.**??** in appendix A.2. We mitigate this issue by fitting correlated models simultaneously with banded ridge regression, and then partitioning the total explained variance across models (Nunez-Elizalde et al., 2019; Dupré la Tour et al.,

2022) to identify the best-performing feature space out of many correlated feature spaces that could independently explain similar amounts of variance.

These results do not tell us about the performance of our features relative to other possible feature spaces that we have not included in the regression. Since our encoding model, based on features from the LAV driving DNN, is only one hypothesis about the representations the brain may use for driving, in future work we plan to compare encoding models based on other algorithms for driving. It would be especially interesting to investigate encoding models that incorporate emerging trends in deep learning for autonomous driving such as world modeling and uncertainty modeling (Chen et al., 2024). Building encoding models using features from alternative driving DNNs will enable us to explore which DNN network architectures and training objectives better reflect the algorithms and representations used by the human brain for driving.

To our knowledge, we have presented the first study that compares brain activity and DNN activations in an interactive, closed-loop, and naturalistic task. Driving requires the agent to continuously perceive the stimulus, make decisions, and produce actions that in turn affects the perceived stimulus. Previous work relating DNNs to the brain have all used perceptual tasks in which the agent could not interact with the stimulus, such as viewing photographs (Yamins et al., 2014), listening to words or music clips (Kell et al., 2018), or categorizing an image (Flesch et al., 2022). Such tasks engage a single sensory system of the brain and reflect only a very narrow subset of natural behavior. Here, we related not only perceptual representations between a neural network and the brain, but also representations for using these perceptual inputs to make plans and produce action outputs.

## 5 CONCLUSION

In this work, we have presented an in-depth comparison of the cognitive representations for driving in humans and in a deep end-to-end neural network for autonomous driving. By directly comparing the representations in a deep learning driving algorithm with humans for the first time, we directly evaluate the alignment between the human brain and the components of a driving network. Our results highlight a striking similarity in the way that driving-relevant representations are organized between humans and the deep neural network. However, we also identified aspects of the network that might be less aligned with humans, including the representation of other vehicles on the road. While driving algorithms do not necessarily need to mimic humans to perform well, humans are still more capable than autonomous vehicles in most environments. Thus, better alignment with human representations may improve autonomous vehicle performance. Furthermore, better alignment with human representations may enable autonomous vehicles to make better inferences about other (human) drivers, an important component of safe and socially acceptable driving. In turn, deep neural networks can provide novel insights about how the brain represents information for solving complex tasks in dynamic environments. Our approach provides a promising framework for understanding how both humans and AI agents solve dynamic, active tasks, both in driving as well as in other domains.

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
