## A   APPENDIX

### A.1   RISK ANALYSIS OF RIDGE REGRESSION UNDER SPARSE RANDOM PROJECTION

In this section, we characterize how a sparse random projection of the data matrix $X$ affects the regression error in ridge regression. Our analysis is based on combining the random projection matrix analysis in Achlioptas (2003) and the ridge regression analysis technique in Lu et al. (2013).

Let $X \in \mathbb{R}^{n \times d}$ be the data matrix containing $n$ i.i.d. samples from a $d$-dimensional independent random variable $x$ such that $d >> n$. $Y \in \mathbb{R}^{n \times 1}$ is the real valued response vector. $\mu \in \mathbb{R}^{n \times 1}$ is the Homoscedastic noise vector with zero mean and common variance $\sigma^2$. Let $\beta \in \mathbb{R}^d$ be the parameter vector, and we have $Y = X\beta + \mu$. We infer the parameter vector $\hat{\beta}_\lambda \in \mathbb{R}^{d \times 1}$ by solving the ridge regression optimization problem:

$$\hat{\beta}_\lambda = \arg \min_{\beta \in \mathbb{R}^{d \times 1}} \frac{1}{n} \|Y - X\beta\|_2^2 + \lambda \|\beta\|_2^2 \tag{1}$$

where $\hat{\beta}_\lambda = (X^\top X + n\lambda I_d)^{-1} X^\top Y$. Since inverting $(X^\top X + n\lambda I_d \in \mathbb{R}^{d \times d}$ is expensive, alternatively we can consider the dual formulation of the ridge regression by first defining its kernel matrix $\mathbf{K} = XX^\top$:

$$\hat{\alpha}_\lambda = \arg \min_{\alpha \in \mathbb{R}^{n \times 1}} \frac{1}{n} \|Y - \mathbf{K}\alpha\|_2^2 + \lambda \alpha^\top \mathbf{K} \alpha \tag{2}$$

where the optimal solution is $\hat{\alpha}_\lambda = (\mathbf{K} + n\lambda I_n)^{-1} Y$ which yields $\hat{\beta}_\lambda = X^\top \hat{\alpha}_\lambda$.

Let $\beta > 0$ and $0 < \epsilon < \frac{3}{2}$. Pick $k \geq \frac{4 + 2\beta}{\epsilon^2/2 - \epsilon^3/3} \log(n)$. Define the projection matrix $\Theta \in \mathbb{R}^{k \times d}$ as in Achlioptas (2003).

$$\Theta_{ij} = \sqrt{\frac{3}{k}} \times \begin{cases} +1 & \text{with probability } 1/6 \\ 0 & \text{with probability } 2/3 \\ -1 & \text{with probability } 1/6 \end{cases} \tag{3}$$

Theorem 1.1 in Achlioptas (2003) shows that, with probability at least $1 - n^{\frac{1}{\beta}}$,

$$(1 - \epsilon)\|u - v\|_2^2 \leq \|\Theta u - \Theta v\|_2^2 \leq (1 + \epsilon)\|u - v\|_2^2, \quad \forall u, v \in \mathbb{R}^d \tag{4}$$

This implies that, with probability at least $1 - n^{\frac{1}{\beta}}$, the singular values of $\Theta$ can be bounded as follows (Eldar & Kutyniok, 2012, Ch.5):

$$(1 - \epsilon) \leq \sigma_{\min}(\Theta) \leq \sigma_{\max}(\Theta) \leq (1 + \epsilon) \tag{5}$$

Furthermore, we can characterize the projected kernel matrix $(X\Theta^\top)(\Theta X^\top)$ with the following result.

**Lemma 1.** *Let $\beta > 0$ and $0 < \epsilon < \frac{3}{2}$. Pick $k \geq \frac{4 + 2\beta}{\epsilon^2/2 - \epsilon^3/3} \log(n)$. Define $\Theta \in \mathbb{R}^{k \times d}$ as in (3), then with probability at least $1 - n^{\frac{1}{\beta}}$,*

$$\|\Theta^\top \Theta - I_d\|_2 \leq \epsilon \tag{6}$$

*Proof.* From Theorem 1.1 in Achlioptas (2003) and the singular value bounds on the random projection matrix in Chapter 5 of Eldar & Kutyniok (2012), we have with probability at least $1 - n^{\frac{1}{\beta}}$,

$$\|\Theta^\top \Theta - I_d\|_2 \leq \sigma_{\max}(\Theta) - 1 \leq \epsilon \tag{7}$$

$\square$

Define $\mathbf{K} := XX^\top$, and $\mathbf{K}_\Theta := (X\Theta^\top)\Theta X^\top$.

**Corollary 1.** *Assuming the same conditions as in Lemma 1, we have, with probability at least $1 - n^{\frac{1}{\beta}}$,*

$$(1 - \epsilon)\mathbf{K} \preceq \mathbf{K}_\Theta \preceq (1 + \epsilon)\mathbf{K} \tag{8}$$

Let $Z = \mathbb{E}_\mu(Y) = X\beta$. Under the fixed design setting (Lu et al., 2013), the risk of a prediction $\hat{Y}$ is $\frac{1}{n}\mathbb{E}_\mu\|\hat{Y} - Z\|_2^2$. For any positive semidefinite matrix $M \in \mathbb{R}^{n \times n}$, let $\hat{Y}_M = X\hat{\beta}_\lambda(M)$ be the prediction obtained by solving the ridge regression problem (2) using $M$ in place of $\mathbf{K}$. It turns out that the risk of this prediction can be expressed using the following function $R$ of $M$ (Lu et al., 2013):

$$R(M) := \frac{\sigma^2}{n}\text{Tr}(M^2(M + n\lambda I_n)^{-2}) + n\lambda^2 Z^\top (M + n\lambda I_n)^{-2} Z \tag{9}$$

Note that the expectation in the risk function is only over the random noise $\mu$.

Similar to Theorem 1 in Lu et al. (2013), we have the following result:

**Proposition 1.** *Let $\beta > 0$ and $0 < \epsilon < \frac{3}{2}$. Pick $k \geq \frac{4+2\beta}{\epsilon^2(1/2-\epsilon/3)}\log(n)$. Consider a random projection matrix $\Theta \in \mathbb{R}^{d \times k}$ defined as in equation 3. With probability at least $1 - n^{\frac{1}{\beta}}$,*

$$R(\mathbf{K}_\Theta) \leq (1 - \epsilon)^{-2} R(\mathbf{K}) \tag{10}$$

*Proof.* For the symmetric positive semi-definite matrix $\mathbf{K}_\Theta$, from Bach (2013), we have that the function

$$B(\mathbf{K}_\Theta) := n\lambda^2 Z^\top (\mathbf{K}_\Theta + n\lambda I_n)^{-2} Z \tag{11}$$

is non-increasing in $\mathbf{K}_\Theta$, and the function

$$V(\mathbf{K}_\Theta) := \frac{\sigma^2}{n}\text{Tr}(\mathbf{K}_\Theta^2(\mathbf{K}_\Theta + n\lambda I_n)^{-2}) \tag{12}$$

is non-decreasing in $\mathbf{K}_\Theta$. Note that $R(\mathbf{K}) = B(\mathbf{K}) + V(\mathbf{K})$. By Corollary 1, we have

$$\begin{aligned}
R(\mathbf{K}_\Theta) &= V(\mathbf{K}_\Theta) + B(\mathbf{K}_\Theta) \\
&\leq V((1 + \epsilon)\mathbf{K}) + B((1 - \epsilon)\mathbf{K}) \\
&\leq (1 + \epsilon)^2 V(\mathbf{K}) + (1 - \epsilon)^{-2} B(\mathbf{K}) \\
&\leq (1 - \epsilon)^{-2} R(\mathbf{K})
\end{aligned} \tag{13}$$

$\square$

Proposition 1 shows that the risk of ridge regression under a sparse random projection can be upper bounded as a function of $\epsilon$ with high probability (at least $1 - n^{\frac{1}{\beta}}$). However, there are still two differences between the methodology in our work and the above analysis. First, we use modified probabilities for the entries of $\Theta$ that enable faster projections at the cost of a small additional approximation error, as described in Li et al. (2006), which is important for efficiently extracting features from a large number of samples. Second, we use banded ridge regression (which is equivalent to multiple-kernel ridge regression (Dupré la Tour et al., 2022)) instead of regular ridge regression. We leave an extension of our analysis to multiple-kernel ridge regression to future work.

## A.2 SUPPLEMENTARY FIGURES

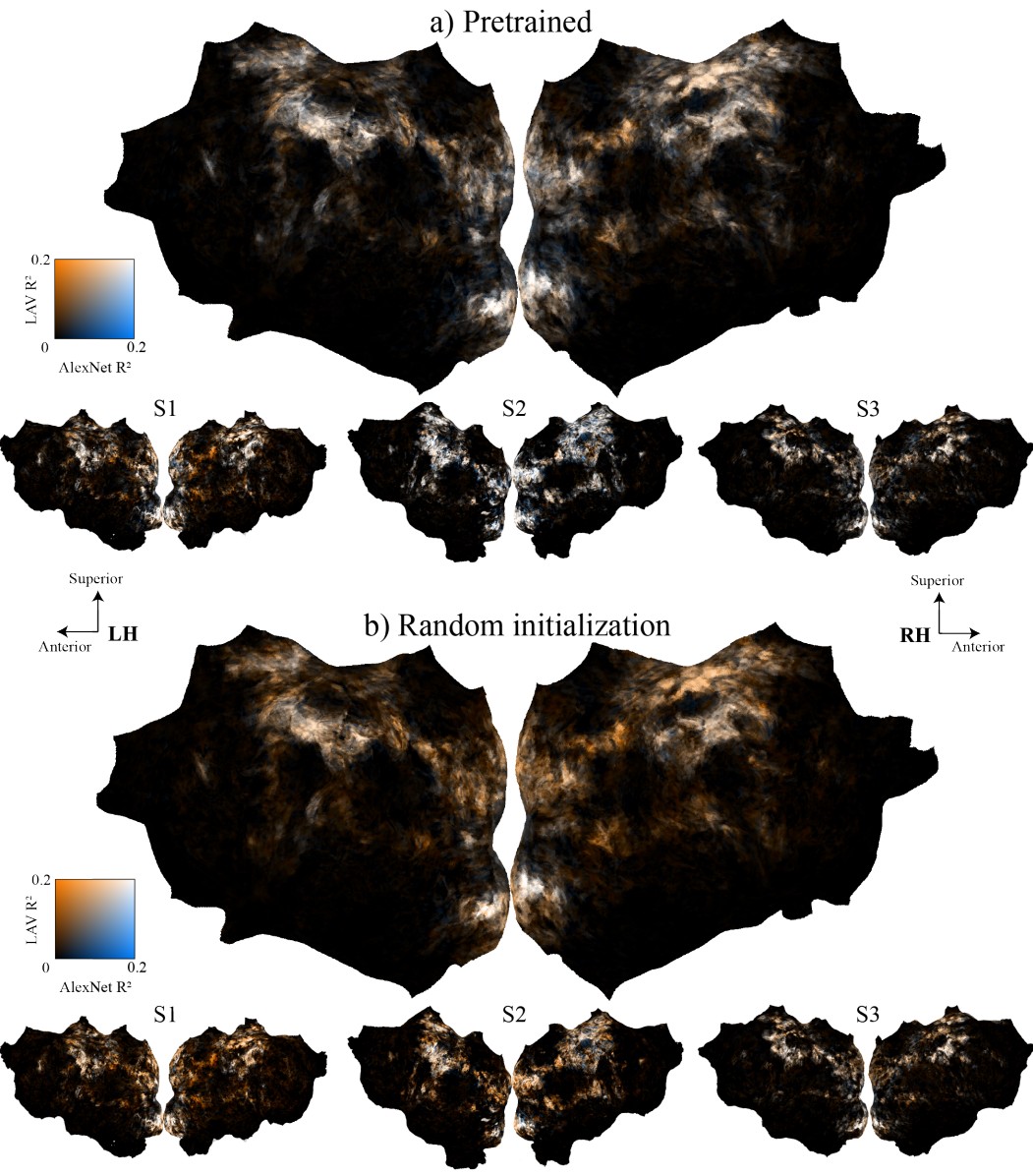

Figure 4: **Features from the LAV DNN explain more variance across the brain than an image classification CNN.** In each subplot, voxels are colored based on the explained variance ($R^2$) of the model derived from LAV DNN activations (orange) and that of a model derived from the activations of an AlexNet CNN. The AlexNet CNN received the same RGB inputs as the LAV DNN, but no lidar inputs or destination waypoint. The four RGB images at each frame were used as separate inputs to AlexNet, and then the activations corresponding to each image were concatenated before performing dimensionality reduction with a sparse random projection with $K = 20000$ components. a): Comparison between LAV and an AlexNet CNN trained on ImageNet (using the best weights in the `torchvision` library) at the group level (top) and for individual subjects (below). LAV explains more variance across the brain, especially in high-level vision areas. b): Comparison between LAV and an untrained AlexNet CNN with random weight initializations at the group level (top) and for individual subjects (below). LAV explains more variance across the brain.

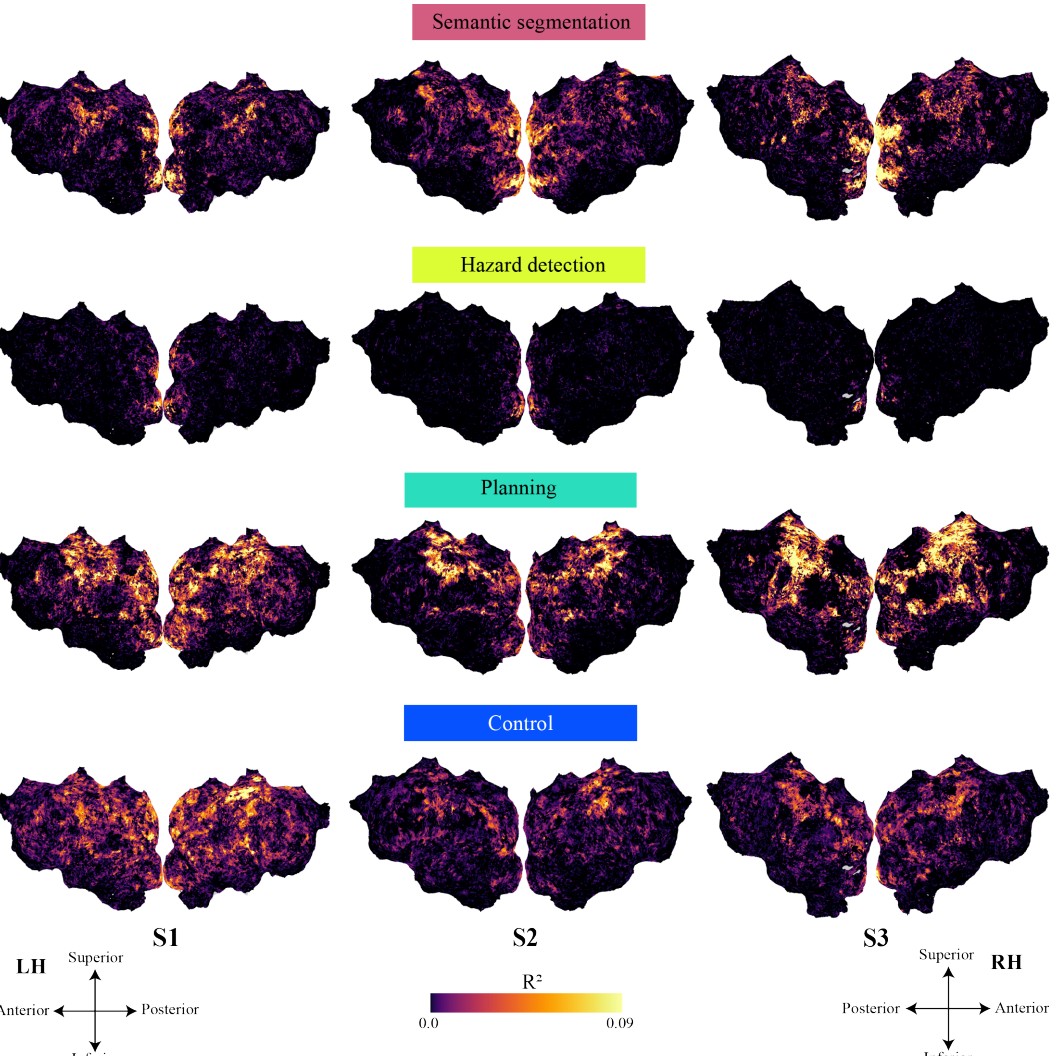

Figure 5: **Subject-level figures corresponding to the group-level figures in Fig.2c**

### A.3 MORE DETAILS ON METHODOLOGY

#### A.3.1 SCANNING PROCEDURE

BOLD activity were acquired on a 3T Siemens Trio with a 32-channel head coil, located at [redacted], with a T2*-weighted gradient-echo EPI sequence customized with a water-excitation radiofrequency pulse to prevent contamination from fat signal (TR = 2.0045 s, echo time = 35 ms, flip angle = 74°, voxel size = 2.24 × 2.24 × 3.5 mm$^3$, field of view = 224 × 224 mm$^2$, matrix size = 100 × 100, and 30 axial slices to cover the entire cortex). Custom personalized headcases (caseforge, Power et al. (2019)) were used to stabilize the head and to reduce motion artifacts. Anatomical data were also collected to reconstruct the cortical surface (three-dimensional T1-weighted MP-RAGE sequence, 1 × 1 × 1 mm$^3$ voxel size and 256 × 212 × 256 mm$^3$ field of view).

Data were collected across multiple scanning sessions. Each session consisted of six 11-minute functional runs. Pilot experiments showed that good voxelwise models could be fit with about two hours of data, so at least two hours of data were recorded from each participant (3 hours each from subjects 1 and 2 and 2 hours from subject 3). Respiration and heart rate were recorded using a

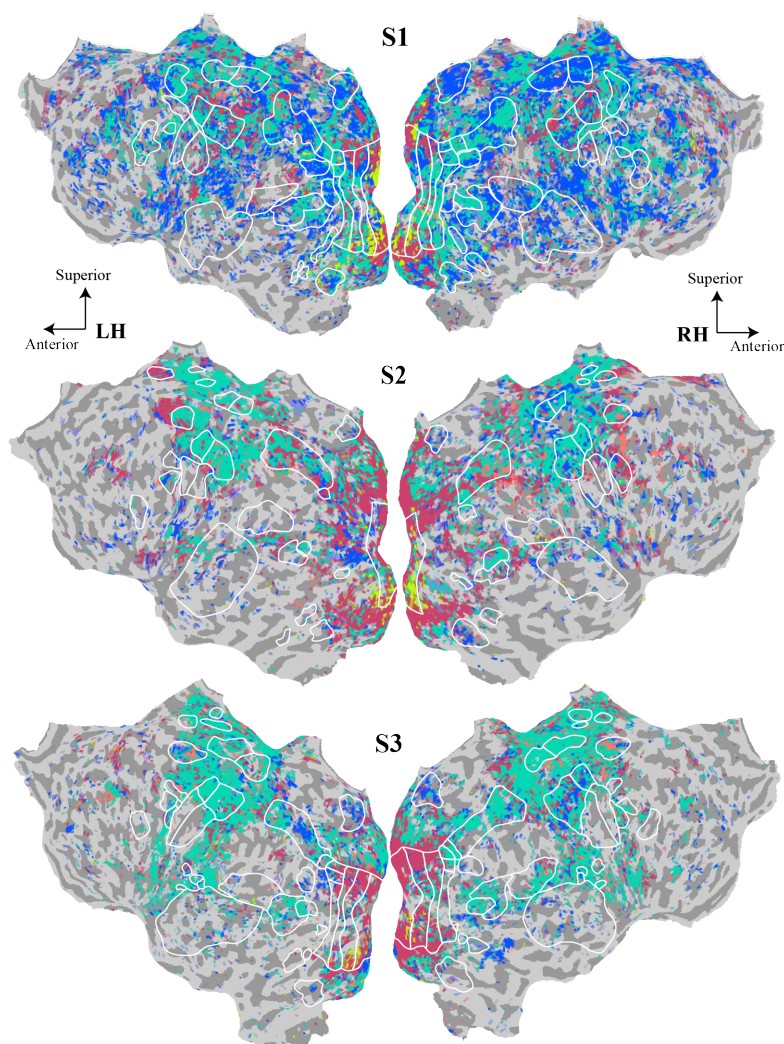

Figure 6: **Subject-level figures corresponding to the group-level plot in Fig.2b**

BIOPAC MP150 system (BIOPAC Systems, Inc.). Gaze location were recorded using an Avotec dark-pupil IR eyetracker at 60 Hz.

### A.3.2 FMRI DATA PREPROCESSING

Each functional run was first motion-corrected using the FMRIB Linear Image Registration Tool from FSL 5.0 (Jenkinson & Smith, 2001; Jenkinson et al., 2002). Next, functional images were unwarped by applying FUGUE from FSL to fieldmaps collected between functional runs. All volumes in the run were then averaged across time to obtain a high-quality template volume. Data across multiple sessions and runs were then registered to the template volume from the first session. Pycortex (Gao et al., 2015) was then used to register the functional data to the anatomical surface. Alignment was checked manually and adjusted as necessary to improve accuracy. Low-frequency voxel response drift was identified using COMPCOR (Behzadi et al., 2007) and removed from the signal. Finally, physiological signals from respiration and heartbeats were regressed out. Voxel activity in each 11-minute run was z-scored separately; that is, within each run, the mean response for each voxel was subtracted and the remaining response was scaled to have unit variance. To remove confounds from the eyetracking calibration sequence and detrending artifacts, the first 35 and last 5 TRs were then discarded from each run.

### A.3.3 LOCALIZERS FOR KNOWN FUNCTIONAL REGIONS

Separately from the functional data collection, five sets of localizer data were collected from each subject. These included a retinotopic localizer used to delineate V1, V2, V3, V4, V3A, V3B, and V7 (Hansen et al., 2007), a MT localizer to delineate the human middle temporal complex (hMT+), a visual category localizer to delineate the fusiform face area (FFA), the extrastriate body area (EBA), the parahippocampal place area (PPA), the occipital place area (OPA), and the retrosplenial cortex (RSC), a motor localizer to delineate the primary motor and somatosensory areas for hands (M1H, S1H), feet (M1F, S1F), and mouth (M1M, S1M), and also the intraparietal sulcus (IPS), frontal eye fields (FEF), the frontal operculum (FO), the superior ventral premotor speech area (sPMv), and Broca's area.

### A.3.4 STATISTICAL TESTS

Permutation tests were used to establish statistical significance for per-voxel encoding model performances. Model predictions were permuted 1,000 times to establish a null distribution for model performance. Significant voxels were then selected at the Bejamini-Hochberg FDR-corrected $p < 0.01$ threshold.

The Moran's $I$ statistic was used to quantify the spatial autocorrelation of the best-performing LAV DNN modules across the brain. The weight matrix was set so that the $i, j$th index was 1 for voxels adjacent to or diagonal from each other in 3D space, and 0 otherwise. The best performing modules at each voxel were permuted 1,000 times to establish a null distribution for the significance of the statistic. A threshold of $p < 0.01$ was used to determine significance.