# OpenReview forum: "Representations in a deep end-to-end driving model predict human brain activity in an active driving task"
_ICLR.cc/2025/Conference — Submitted to ICLR 2025_

### Official Review · Reviewer_1Y8c · 2024-10-23

**Soundness:** 2
**Presentation:** 3
**Contribution:** 2
**Rating:** 5
**Confidence:** 3

**Summary:**

This paper focuses on the alignment between deep learning models and brain activity. Unlike previous studies, which examine the alignment of visual or language models with brain activity, this work explores a deep learning model for autonomous driving. Specifically, the paper utilizes the LAV model, which has clearly separated functional modules, including semantic segmentation, bird's-eye view perception, planning, trajectory prediction, and hazard detection. The outputs from each module demonstrate varying predictive capacities across functionally distinct brain regions.

**Strengths:**

The topic comparing the brain activity to a autonomous driving model is quite new to the field which can be insightful for understanding the brain activity during planing, decision making. The submission collects the data with this new system is a good start point for the following research.

**Weaknesses:**

Though the topic is new, mapping an autonomous driving model with distinct functional modules to different brain regions is promising, but the current results are not yet strong enough. For instance, the control module outputs show high predictive ability across multiple brain regions, it would be beneficial if the authors could demonstrate whether these regions are consistent across random sees, subjects and providing some statistical significance measure.

Additional concerns are as follows:

The authors performed regression analysis to align LAV model outputs with brain activity. It would be helpful to clarify whether the observed distinct predictive abilities are specific to the LAV model or if they generalize across other autonomous driving models, such as that proposed by Li et al., 2024 [1].

[1] Li et al., 2024, https://arxiv.org/html/2406.08481v1.

Predictive ability is a coarse measure, as it only indicates that the variability in model outputs aligns with the variability in brain activity. This makes it difficult to draw conclusions such as "representations learned by the driving DNN may be similar to those used by the human brain." The authors could explore additional metrics beyond regression fitting to better align brain activity, such as fMRI, with artificial neural networks. A discussion on the impact of metrics on alignment-related conclusions would also be beneficial [2].

[2] Soni et al., 2024, https://www.biorxiv.org/content/10.1101/2024.08.07.607035v1.full.pdf.

While the topic is interesting, current technical contribution is not very significant.

**Questions:**

1. What is the variability across subjects? When the authors mentioning the group-level performance, does that mean average across subjects?

2. How does the random projection matrix affect the results?

3. Is there any statistical measure quantifying the significance of the better predictive ability of one brain region compared to other regions?

---

> ### Author Response · Authors · 2024-11-23
> **Response to reviewer 1Y8c (1/2)**
>
> We would like to thank the reviewer for their helpful comments, which have been valuable for improving the paper, and for noting that the topic of the paper is novel and interesting. We first provide a brief summary of our changes, then address the reviewer’s points in more detail below. We have performed new baseline comparison experiments in appendix, which we have added to appendix A.2. These experiments show that the LAV DNN features are able to explain more variance in brain activity than those of a standard CNN trained on image classification, especially in high-level vision areas. We also added statistical tests for voxel encoding model performance and for the distribution of best-performing modules across the cortex, which establish the statistical significance of these results. Finally, in order to address questions about the methodology, we have revised section 3 and added appendix A.3 to clarify details about the voxelwise modeling framework as well as why this framework is rigorous and suitable for our analysis.
>
> The reviewer expresses three main concerns. First, they suggest additional analysis would better support the results. To strengthen our analysis, we have provided statistical significance measures for per-voxel $R^2$ scores, as well as for the non-random partitioning of explained variance by each LAV module across different voxels. We have also provided figures for the individual subjects in the supplementary. We find that there is in fact good consistency between the explained variance of the LAV modules across subjects, especially for the semantic segmentation, brake, planning, and control modules. Furthermore, the prediction and BEV perception modules predict variance in functionally similar regions across subjects as shown in figure 3, even though these regions do not project to the same exact locations in the group-level space.
>
> Second, the reviewer wonders whether the results generalize to other driving DNNs. We agree that incorporating comparisons to other driving models to improve our understanding of whether these models converge on similar levels of brain alignment and functional organization is an exciting area. Because running experiments with other models which expect different inputs (e.g. types and positions of cameras) requires rendering different data in addition to repeating regressions for each subject, this isn’t possible for us to address during the rebuttal period, but is an interesting direction for future work. However, we have added new baseline comparisons with CNN models that support the strength of the LAV DNN features at explaining brain activity, and that the benefit of LAV features over CNN features is strongest in specific functional areas.
>
> Third, the reviewer questions the choice of VM and linear predictivity as the metric for quantifying alignment between the brain and a DNN. The metrics explored in Soni et al and frequently used in studies attempting to quantify brain alignment are population-based metrics that return a single value for the alignment between two sets of features (the entire DNN and the entire brain, or specific sub-regions of either). Therefore, they do not allow for the per-voxel analysis that we focus on in this paper, and a comparison with other methods would require major changes to our analysis that is beyond the scope of this paper. We would also like to note that VM is the most powerful method for analyzing complex brain activity recorded during naturalistic tasks, and has been validated in multiple studies [Huth et al., 2012, Huth et al., 2016, Nishimoto et al., 2011, Cukur et al., 2013, Deniz et al., 2019]. It is the only method that explicitly models the timeseries recorded from the brain, and is thus uniquely suited for analyzing data from continuous tasks. Finally, the regression methods underlying VM are statistically rigorous [Dupre la Tour et al., 2022, Nunez-Elizalde et al., 2019] and are drawn from solid mathematical foundations in linear regression.

---

> > ### Author Response · Authors · 2024-11-23
> > **Response to reviewer 1Y8c (2/2)**
> >
> > **Response to questions:**
> > 1. Question 1 is about the variability between subjects and the group-level performance methodology. We have added additional figures to appendix A.2 to show results for individual subjects (please refer to response point 1 above). The group-level performance indeed refers to the average model performance across subjects. More specifically, we use the FreeSurfer fsaverage surface as the group template. For each subject, we use FreeSurfer’s surf2surf to compute a mapping from each subject’s cortical surface to the template surface. This projection is based on warping the topology of the subject’s cortical surface to best match the shared topology. Model performances from all subjects are projected to and then averaged on the fsaverage surface. We have included additional information in appendix A.3 to clarify these details of our methodology.
> >
> > 2. Question 2 is about the impact of the sparse random projection matrix. As the number of random projection components increases, the choice of random projection matrix should have decreasing influence on the contents of the features by the Johnson-Lindenstruss lemma, and therefore on the performance of ridge regression by the proof in appendix A.1. We verified that for our selected number of components (20,000 per module) the choice of random projection matrix has minimal effect on our results in practice by repeating our entire feature extraction and regression pipeline with a second random matrix and observing minimal differences.
> >
> > 3. Question 3 is about a statistical test for model predictive ability. As noted in response point 1 above, we have added a statistical test for this and used it to limit the voxels shown in figure 2b to those with statistically significant model performance.
> >
> > **References**
> >
> > Huth, A.G., Nishimoto, S., Vu, A.T. and Gallant, J.L., 2012. A continuous semantic space describes the representation of thousands of object and action categories across the human brain. Neuron, 76(6), pp.1210-1224.
> >
> > Huth, A.G., De Heer, W.A., Griffiths, T.L., Theunissen, F.E. and Gallant, J.L., 2016. Natural speech reveals the semantic maps that tile human cerebral cortex. Nature, 532(7600), pp.453-458.
> >
> > Nishimoto, S., Vu, A.T., Naselaris, T., Benjamini, Y., Yu, B. and Gallant, J.L., 2011. Reconstructing visual experiences from brain activity evoked by natural movies. Current biology, 21(19), pp.1641-1646.
> >
> > Çukur, T., Nishimoto, S., Huth, A.G. and Gallant, J.L., 2013. Attention during natural vision warps semantic representation across the human brain. Nature neuroscience, 16(6), pp.763-770.
> >
> > Deniz, F., Nunez-Elizalde, A.O., Huth, A.G. and Gallant, J.L., 2019. The representation of semantic information across human cerebral cortex during listening versus reading is invariant to stimulus modality. Journal of Neuroscience, 39(39), pp.7722-7736.
> >
> > Dupre La Tour, T.D., Eickenberg, M., Nunez-Elizalde, A.O. and Gallant, J.L., 2022. Feature-space selection with banded ridge regression. NeuroImage, 264, p.119728.
> >
> > Nunez-Elizalde, A.O., Huth, A.G. and Gallant, J.L., 2019. Voxelwise encoding models with non-spherical multivariate normal priors. Neuroimage, 197, pp.482-492.

---

> > > ### Comment · Reviewer_1Y8c · 2024-11-28
> > >
> > > Thanks authors for their detailed responses.
> > >
> > > Based on additional single subject results shared by authors basically, Fig.3 and Fig. 4, it seems the activation pattern is not very consistent across subjects, or even left vs. right hemisphere (e.g., OFA, EBA explained variance patters are different across subjects). I think it would be hard to draw a strong connection between DNN partition tasks similar to human brain with these inconsistence across subjects.
> > >
> > > For the sparse random projection matrix, the proof shared by author does not exactly match the way random project matrix used in main paper right? Could you share the quantity of the observed 'minimal differences' regarding your response to question 2?

---

> > > > ### Author Response · Authors · 2024-12-04
> > > >
> > > > Thank you for your response. Yes, you're correct about the random projection matrix. Quantitatively, repeating the analysis with a new random projection matrix yielded an average absolute difference of 3e-3 to 6e-3 in the model $R^2$ in significantly predicted voxels (3e-3 in two subjects, 6e-3 in one subject). Also, 13-16% of significant voxels have a different most-predictive LAV module with a different random projection. However, qualitatively, we find that these differences do not change the overall pattern of the most predictive modules across the brain. Please see [this figure](https://figshare.com/s/c93746fc65e4485a993f) for an example of three separate regression results in subject 3: the first is the original, the second has the same random projection but a separate run of the banded ridge regression pipeline, and the third has a different random projection as well as a separate regression.

---

### Official Review · Reviewer_UjYj · 2024-10-31

**Soundness:** 2
**Presentation:** 2
**Contribution:** 2
**Rating:** 5
**Confidence:** 3

**Summary:**

This paper studied how the representations in a deep learning model for autonomous driving can predict human brain responses in an interactive close-loop driving task. They recorded human subjects' brain activities using fMRI while they engaged in a driving simulation task. They extracted activations from artificial neurons in the deep network model receiving stimuli similar to human subjects and used these activations to regress against brain activities. They found that overall, the model explains variances of brain responses across many brain regions in held-out data. They further investigated how different modules in the deep learning model, such as semantic segmentation, planning, hazard detection, and control, explain different parts of the brain responses the best. They found that semantic segmentation and hazard detection modules best predict the visual areas, the planning module best explains variance in the sensorimotor areas and IPS, and the control module is similar to the planning module and, in addition, explains variance in RSC and PPA.

**Strengths:**

This paper studies human neural activities in a complex interactive driving task. It investigates to what extent a functional model of driving—and its different submodules—explains/predicts different parts of the neural data. Many works in the past investigated how deep neural network models align and/or predict neural responses, but most previous studies focused on perception, reasoning/planning, or control separately, and the tasks were usually much simpler. This work studies driving, a complex interactive behavior involving perception, planning, and control. Going from simple, passive tasks to complex, multifaceted tasks has significant originality. Meanwhile, developing capable computational models and comparing different facets of the model to the brain involves a lot of hard work and innovation in methodology, and this work made progress in that direction. The finding that different submodules of the LAV model explain variance in brain responses in different regions is a novel finding and invites further studies to understand the exact functional roles of different brain regions during a complex task such as driving.

**Weaknesses:**

While the task, model, and analysis methods are novel, it is hard to know what we have learned scientifically from the analysis, mainly due to a lack of control experiments and alternative models. I see the central claims in this paper as the following two points.

1. encoding models for DNN activations explain significant amounts of variance in brain activity across many regions of the brain
2. each functional module in the DNN explains brain activity in a distinct network of functional regions in the brain, ..., suggesting that both the DNN and the human brain may partition the task in a similar manner.

Claim 1 is not novel since it is generally expected that a DNN model can account for variance in neural response, especially when these models are trained to perform the same task. Even randomly initialized DNN models can explain some variance in the brain. Given that, it is essential to see how well the LAV model explains variance compared to other models. Does LAV predict brain activities better in a particular region, or does it predict activities in a broader range of areas? For example, the author can compare the LAV model to those non-DNN models studied by Strong et al., 2024., and it would be helpful to have more DNN control models, such as a CNN trained on ImageNet classification or a randomly initialized CNN model.

While this paper did show that different submodules of LAV explain variance in different brain regions, the claim that the brain and LAV partition the task in a similar manner is only poorly supported. This is primarily due to a lack of clarity on what "partitioned similarly" means. From the presented data, the semantic segmentation and hazard detection modules explain the neural responses in the visual areas. The planning and control modules explain a largely overlapping set of brain regions. These results suggest that the functions performed by these modules are not as clearly segregated in the brain as in the LAV model. Establishing a clear metric to assess whether the brain exhibits a similar functional partitioning as the tested model would be beneficial. This could involve developing a measure of the degree of functional segregation in the model that aligns with brain regions. Adding alternative models or control models would certainly help. For example, there might be a hypothetical model A, whose sub-modules predict all brain regions equally well. Then, it is acceptable to conclude that the LAV model partition is more brain-like than model A.

Additionally, while this paper mainly focuses on analyzing the neural data, it does not provide any behavioral results. It is hard to see the model as a good model of the brain if it does not perform the task well or does not match human behavior well. It would be helpful to see how well the LAV model is aligned with humans behaviorally. For example, the navigation decisions between the LAV model and human subjects can be compared when given the same simulator inputs.

Reference:

Strong, C., Stocking, K., Li, J., Zhang, T., Gallant, J. and Tomlin, C., 2024, June. A framework for evaluating human driver models using neuroimaging. In 6th Annual Learning for Dynamics & Control Conference (pp. 1565-1578). PMLR.

**Questions:**

1. How well does the LAV model explain brain responses compared to non-DNN baseline models, such as those studied in Strong et al., 2024.
2. How well does the LAV model explain brain responses compared to other DNN models? For example, some basic baseline DNN models, such as an ImageNet-trained CNN. Or some alternative driving DNN models.
3. How can we more rigorously measure whether a computational model and the brain partition the task similarly?
4. How well does the behavior from the computational model align with human behavior?

---

> ### Author Response · Authors · 2024-11-23
> **Response to reviewer UjYj (1/2)**
>
> We would like to thank the reviewer for their helpful comments, which have been valuable for improving the paper. We first provide a brief summary of our changes, then address the reviewer’s points in more detail below. We have performed new baseline comparison experiments in appendix, which we have added to appendix A.2. These experiments show that the LAV DNN features are able to explain more variance in brain activity than those of a standard CNN trained on image classification, especially in high-level vision areas. We also added statistical tests for voxel encoding model performance and for the distribution of best-performing modules across the cortex, which establish the statistical significance of these results. Finally, in order to address questions about the methodology, we have revised section 3 and added appendix A.3 to clarify details about the voxelwise modeling framework as well as why this framework is rigorous and suitable for our analysis.
>
> The reviewer raises three main concerns. First, the reviewer suggested that comparing the LAV model against appropriate driving model baselines would strengthen the results. Unfortunately it isn’t possible to do a direct comparison with the models in [Strong et al., 2024], as these models only work when there are no intersections or turning behavior. Furthermore, as these models do not handle the state estimation problem of predicting the positions and dynamics of other vehicles from perceptual inputs, they are at a significant disadvantage compared to a DNN that processes image-based input. Finding ways to make meaningful comparisons in spite of these challenges is an exciting direction for future work.
>
> We appreciate the suggestion of image classification models as an appropriate baseline. We have added comparisons with two new models: one derived from the activations of a randomly initialized CNN, and one from the activations of a CNN trained on ImageNet image classification. Our model performs better than both baseline models, especially in higher-level vision areas. This supports our experimental design capturing explainable brain activity beyond what can be explained with image processing features alone, and that the LAV DNN is in fact able to explain some of this additional variance. Results and more details about these experiments can be found in appendix A.2.
>
> Second, the reviewer wanted stronger support for the claim that LAV and the brain partition features in a similar way. We agree that while it is not straightforward to evaluate the similarity in partitioning, it could nonetheless be evaluated both quantitatively and qualitatively. First, we highlight that encoding models are fit to each voxel independently; the modelling process contains no inductive bias that would encourage models for spatially proximal voxels to have similar partitioning of variance across the different LAV modules. Nevertheless, the best-performing LAV module for each voxel has quantitatively a non-random spatial distribution across the brain. To show that this pattern is statistically significant, we have added a statistical test based on the Moran’s I measure of spatial autocorrelation that finds a p-value of < 0.01 in all three subjects (please see appendix A.3 for more details). This non-random distribution suggests that each model maps to a specific network of functional regions in the brain. Second, the partitioning can be qualitatively evaluated by comparing the LAV module function with the known functional properties of the brain regions to which it is mapped. For example, the semantic segmentation and brake modules, which process RGB images, are the best-performing modules in the visual cortex but are outperformed by planning and control modules in sensorimotor regions. This functional similarity between the LAV module and corresponding brain regions during the same task suggest that they mediate the same aspects of the task. Finally, we agree with the limitation that we have evaluated only a single driving DNN, and comparison of alignment across DNN architectures will be a key direction for future work.

---

> > ### Author Response · Authors · 2024-11-23
> > **Response to reviewer UjYj (2/2)**
> >
> > Third, the reviewer suggested that a good model for human driving should explain human activity at a behavioral level in addition to a cognitive one, and asked about the behavioral match between LAV and the human subjects. LAV is trained to imitate the CARLA expert driving agent rather than human driving behavior. This means that even though it exhibits strong performance in completing driving routes while avoiding safety and traffic rule violations, it is a poor fit for the human subjects behaviorally. Qualitatively, we find that the LAV agent drives much slower and is prone to stopping more frequently than the human subjects. However, we hypothesize that similar representations of the environment and other agents may underlie very different driving styles, which is supported by the ability of the our model to explain brain variance. Attempting to also obtain a better behavioral fit, e.g. by fine-tuning the model on the driving trajectories of individual subjects, is an interesting direction for future work.
> >
> > **Response to questions:**
> >
> > Please see the responses above.

---

> > > ### Comment · Reviewer_UjYj · 2024-12-03
> > >
> > > Thank you for the response!
> > >
> > > - First, I really appreciate that the author added the ImageNet-trained and random AlexNet models as baselines for comparison. These new results show that the LAV model predicts brain responses better than the ImageNet AlexNet model. This partly addresses my previous concerns, but not fully. The AlexNet model is only a weak baseline model in predicting neural responses. Although, in general, the LAV model may explain variance in the brain better. But, from the figure in Appendix A.2, the fact that this weak CNN baseline can explain neural response equally well as LAV in many of the brain regions (white color) and the fact that this vision-only model explains a lot of the regions that correspond to the planning module in Figure 2, makes me think that it is still too early to draw some definitive conclusions here. I think this paper could still be strengthened by comparing the LAV with more baseline models, such as more performant and more modern models.
> > >
> > > - Second, my concern about the claim that LAV and the brain are "partitioned similarly" remains. While the authors mentioned in their rebuttal that the best-performing LAV module for each voxel exhibits a non-random spatial distribution across the brain, this represents only a small step toward addressing this concern. I believe a more rigorous and quantifiable measure of “partitioned similarly,” along with comparisons to additional driving models, is necessary for the authors to substantiate claims about which model aligns more closely with the brain’s partitioning. For instance, in the example that the author gave, the semantic segmentation and brake modules map to overlapping brain regions in the visual cortex, while these two modules are separated in the LAV model. This could indicate that the brain is not partitioned in a way like that of the LAV model.
> > >
> > > For the reasons given above, I will maintain my score for now.

---

### Official Review · Reviewer_map4 · 2024-11-02

**Soundness:** 3
**Presentation:** 3
**Contribution:** 3
**Rating:** 5
**Confidence:** 5

**Summary:**

In this article, the authors present an interesting attempt of aligning the auto-driving neural network with the human brains scanned when driving. This experiment is a new design and allows for the exploration of new topics in the field.

**Strengths:**

1. The dataset is quite new.
2. The visualization is clear and neat.

**Weaknesses:**

1. The sample size is relatively small. I understand the difficulty here and I guess the whole collection is still in the early stage?
2. The goodness of mapping is not well evaluated.
3. The comparison with other methods and infrastructure is missing.

**Questions:**

1. The authors may include more information about data processing and mapping in the supplement.
2. More details about the quantitative analysis could be included.
3. The authors may include some comparison with the non-specific encoding models.
4. How do you align the driving pattern between human and AI? Are they aligned with the same frame or action? As the performance is measured by R^2, is it selective to the current BOLD? What's the difference if you map it to a resting or passive natural stimulus? To what extend is the signal driven by the movement? Some related work could be helpful for the comparison and argument here about the selection and representation, such as:
  [1] https://www.nature.com/articles/s41467-024-53147-y
  [2] https://www.nature.com/articles/s42256-023-00753-y?fromPaywallRec=false
  [3] https://www.sciencedirect.com/science/article/pii/S2095927324001373
  [4] https://openaccess.thecvf.com/content/CVPR2024/html/Yang_Brain_Decodes_Deep_Nets_CVPR_2024_paper.html

---

> ### Author Response · Authors · 2024-11-23
> **Response to reviewer map4 (1/2)**
>
> We would like to thank the reviewer for their helpful comments, which have been valuable for improving the paper. We first provide a brief summary of our changes, then address the reviewer’s points in more detail below. We have performed new baseline comparison experiments in appendix, which we have added to appendix A.2. These experiments show that the LAV DNN features are able to explain more variance in brain activity than those of a standard CNN trained on image classification, especially in high-level vision areas. We also added statistical tests for voxel encoding model performance and for the distribution of best-performing modules across the cortex, which establish the statistical significance of these results. Finally, in order to address questions about the methodology, we have revised section 3 and added appendix A.3 to clarify details about the voxelwise modeling framework as well as why this framework is rigorous and suitable for our analysis.
>
> The reviewer raises three main concerns. First, the reviewer is concerned that a pool of three subjects is insufficient for drawing statistically sound conclusions. Here we would like to clarify the n in the conceptual framework underlying our analyses and demonstrate that three subjects is in fact sufficient. The small-n concern expressed by the reviewer reflects the classical psychology experiment framework, in which results from a large number of subjects are averaged to draw a group-level conclusion. Neuroimaging experiments under this framework would therefore need to collect data from a large number of subjects, and because of practical limitations, this necessitates collecting less data per subject (typically on the order of one hour per subject). This framework then seeks to create a single model, with particular parameters, for all subjects. However, because of individual differences in anatomy, cognitive strategies, the high-dimensionality of the brain, and the small amount of data collected per subject, these group-level models rarely provide good descriptions of individual subjects and thus the models are of limited use.
>
> Our study instead follows the framework found in psychophysics and neurophysiology (particularly in non-human primates (NHP)) [Asaad et al., 2024]. In this framework, the n is not the number of subjects, but rather the amount of data collected per subject. In our study, we collected 2-3 hours of data per subject in this experiment, and also an additional 5-6 hours of anatomical and functional localizer data that enabled us to reconstruct the cortical surface and delineate known functional regions. This large amount of data from each individual subject is divided into train, validation, and test sets, and models are fit, cross-validated, and tested within each individual subject. Rather than seeking a particular instantiation of a model with particular parameters to apply to all subjects, this framework demonstrates that a particular architecture of model, with possibly different parameters per subject that can account for the idiosyncracies of each subject, can be used to accurately explain the data in all subjects. In other words, the models are fit and statistically tested within each subject, and each subject is in fact a full replication of the experiment [Asaad et al., 2024].
>
> Indeed, studies from NHP neurophysiology and psychophysics under this framework have routinely used as few as two subjects to reveal fundamental insights into the functions of the brain. Neuroimaging studies with small n-in-subjects have also produced robust models of the human brain in complex, naturalistic tasks. Thus, we have in fact provided sufficient data to prevent overfitting and also replicate this experiment. The reviewer commented that we did not consider this in the text of the manuscript, but we note that this n-in-subjects and n-in-data contrast is a philosophical difference between standard practices across fields and is beyond the scope of this paper.
>
> Second, the reviewer noted that the paper would benefit from more analysis of the model performance. We have added a statistical test for the encoding model performance. This test establishes the statistical significance threshold for each voxel, and we have updated figure 2(b) so that only voxels with p-values over the threshold are shown. We have also added a statistical test for the non-random distribution of best-performing modules across the cortex which shows that this is statistically significant in all three subjects. Details for both tests can be found in appendix A.3.

---

> > ### Author Response · Authors · 2024-11-23
> > **Response to reviewer map4 (2/2)**
> >
> > Third, the reviewer also noted that the paper would benefit from comparisons with appropriate baselines. We have added comparisons with two new models: one derived from the activations of a randomly initialized CNN, and one from the activations of a CNN trained on ImageNet image classification. Our model performs better than both baseline models, especially in higher-level vision areas. This supports our experimental design capturing explainable brain activity beyond what can be explained with image processing features alone, and that the LAV DNN is in fact able to explain some of this additional variance. Results and more details about these experiments can be found in appendix A.2.
> >
> > **Response to questions:**
> >
> > 1. Question 1 is about information about data processing and mapping. We have added a new appendix section on fMRI data preprocessing and mapping. Please refer to appendix A.3 for more information.
> >
> > 2. Please refer to response point 2 above for more details about quantitative analysis.
> >
> > 3. Please refer to response point 3 above for details about new baseline experiments with image classification models that are not specific to driving.
> >
> > 4. The reviewer asked us to clarify how the brain activity and DNN activations are aligned during modelling. Thank you for bringing up these important clarifications. We have updated the material in section 3 to improve the clarity. Our method is based on aligning the inputs to the human subjects and the LAV model. Therefore, for each frame that the human driver sees while completing the interactive driving task, we generate a corresponding set of inputs (RGB images and lidar) to use as inputs to the driving model. Then, we can compare the responses of the human brain activity and driving model activity to matching sets of inputs that correspond to the same state of the environment.
> >
> > The reviewer asks whether the $R^2$ is selective to the current BOLD. We would like to clarify that the $R^2$ is computed separately for each voxel over the entire time series of the test data. Indeed, it does not make sense to compute an $R^2$ value for a single time point.
> >
> > The reviewer also asks whether using a passive task or a resting task would make a difference in the results. To this point, we would expect the results to be very different to the point that it would not be a fair or valid comparison for the results from this task. The brain is a nonlinear system, and changing tasks causes significant changes in neural activity. Perhaps most relevant to this comment, it has been demonstrated that navigation tuning is highly dependent on an active navigation task: place cells in the rodent hippocampus remap between active and passive locomotion in the same environment [Song et al, 2005] and landmark-selective cells in the retrosplenial cortex only display robust landmark selectivity when the animal is actively moving and performing a navigation task. Thus, the brain operates in a different regime in passive and resting tasks, and it is unclear what such a comparison would provide scientifically for the purposes of relating artificial systems and the brain during driving.
> >
> > Finally, the reviewer also asks about the extent to which the signal is driven by movement. We had used custom 3D printed headcases to stabilize the heads of subjects during scanning. Headcases have been demonstrated to be effective at minimizing movement [Power et al. 2019]. Empirically we had also found the motion parameters to be comparable to those obtained from passive movie-watching tasks. Furthermore, the brain images were motion-corrected during preprocessing prior to modelling. This information has been added to the manuscript in the new appendix section A.3.
> >
> > **References:**
> >
> > Asaad, W.F. and Sheth, S.A., 2024. What’s the n? On sample size vs. subject number for brain-behavior neurophysiology and neuromodulation. Neuron.
> >
> > Power, Jonathan D., et al. "Customized head molds reduce motion during resting state fMRI scans." NeuroImage 189 (2019): 141-149.
> >
> > Song, Eun Young, et al. "Role of active movement in place‐specific firing of hippocampal neurons." Hippocampus 15.1 (2005): 8-17.

---

> > > ### Comment · Reviewer_map4 · 2024-11-25
> > > **Response to the rebuttal**
> > >
> > > Thank you for your detailed explanation. However, the concerns on the number of subjects and comparison remain. I would say that I cannot increase my score. Also, given the length and focus of ICLR, I think this paper is more appropriate for a general journal where more detailed analysis and demonstration would be possible and feasible.

---

### Official Review · Reviewer_csZb · 2024-11-02

**Soundness:** 2
**Presentation:** 2
**Contribution:** 2
**Rating:** 3
**Confidence:** 2

**Summary:**

This paper presents a comparison between human brain activity, measured through functional magnetic resonance imaging (fMRI), and activations within deep neural networks (DNNs) during an active taxi-driving task in a naturalistic simulated environment. The study aims to enhance our understanding of the similarities and differences between human cognition and current deep-learning methods in active, closed-loop tasks such as driving.

**Strengths:**

The method is straightforward and easy to understand.

**Weaknesses:**

This paper focuses on application rather than theoretical innovation. Here are a few questions and considerations regarding the methodology:

The sample size is limited to only three subjects. Is this sufficient to establish a reliable confidence level in the findings?

Why was a deep neural network (DNN) chosen over alternative models? Would other models potentially offer comparable or better insights?

The rationale for using the selected model, such as the VM model, remains unclear. Could you clarify the insights driving this choice?

What methods were employed to assess the credibility and robustness of the model? How can we be confident in its generalizability and accuracy?

**Questions:**

Please refer to Weaknesses.

---

> ### Author Response · Authors · 2024-11-23
> **Response to reviewer csZb (1/2)**
>
> We would like to thank the reviewer for their helpful comments, which have been valuable for improving the paper. We first provide a brief summary of our changes, then address the reviewer’s points in more detail below. We have performed new baseline comparison experiments in appendix, which we have added to appendix A.2. These experiments show that the LAV DNN features are able to explain more variance in brain activity than those of a standard CNN trained on image classification, especially in high-level vision areas. We also added statistical tests for voxel encoding model performance and for the distribution of best-performing modules across the cortex, which establish the statistical significance of these results. Finally, in order to address questions about the methodology, we have revised section 3 and added appendix A.3 to clarify details about the voxelwise modeling framework as well as why this framework is rigorous and suitable for our analysis.
>
> The reviewer expresses four main concerns. First, the reviewer is concerned that a pool of three subjects is insufficient for drawing statistically sound conclusions. Here we would like to clarify the n in the conceptual framework underlying our analyses and demonstrate that three subjects is in fact sufficient. The small-n concern expressed by the reviewer reflects the classical psychology experiment framework, in which results from a large number of subjects are averaged to draw a group-level conclusion.
> Neuroimaging experiments under this framework would therefore need to collect data from a large number of subjects, and because of practical limitations, this necessitates collecting less data per subject (typically on the order of one hour per subject). This framework then seeks to create a single model, with particular parameters, for all subjects. However, because of individual differences in anatomy, cognitive strategies, the high-dimensionality of the brain, and the small amount of data collected per subject, these group-level models rarely provide good descriptions of individual subjects and thus the models are of limited use.
>
> Our study instead follows the framework found in psychophysics and neurophysiology (particularly in non-human primates (NHP)) [Asaad et al., 2024]. In this framework, the n is not the number of subjects, but rather the amount of data collected per subject. In our study, we collected 2-3 hours of data per subject in this experiment, and also an additional 5-6 hours of anatomical and functional localizer data that enabled us to reconstruct the cortical surface and delineate known functional regions. This large amount of data from each individual subject is divided into train, validation, and test sets, and models are fit, cross-validated, and tested within each individual subject. Rather than seeking a particular instantiation of a model with particular parameters to apply to all subjects, this framework demonstrates that a particular architecture of model, with possibly different parameters per subject that can account for the idiosyncracies of each subject, can be used to accurately explain the data in all subjects. In other words, the models are fit and statistically tested within each subject, and each subject is in fact a full replication of the experiment [Asaad et al., 2024].
>
> Indeed, studies from NHP neurophysiology and psychophysics under this framework have routinely used as few as two subjects to reveal fundamental insights into the functions of the brain. Neuroimaging studies with small n-in-subjects have also produced robust models of the human brain in complex, naturalistic tasks. Thus, we have in fact provided sufficient data to prevent overfitting and also replicate this experiment.
>
> Second, the reviewer asks why we chose to examine a DNN model over other possible models. There has been one prior work on non-DNN models for driving, which studies algorithms for generating speed and acceleration based on the dynamics and predicted behavior of the vehicle in front [Strong et al., 2024]. However, these dynamical models assume knowledge about the state of the vehicle and environment, and require external processes to provide these state parameters. We note that fully autonomous driving pipelines typically contain at least some DNN components, because only DNNs can reliably estimate the state of the environment from sensor inputs. The human brain drives in an end-to-end manner, and so we believe driving models that make use of DNNs are most appropriate when trying to draw connections to the brain activity of human drivers.

---

> > ### Author Response · Authors · 2024-11-23
> > **Response to reviewer csZb (2/2)**
> >
> > Third, the reviewer asks about the rationale for using voxelwise modelling. Here, we note that in neuroimaging, VM is the most powerful method for analyzing complex brain activity recorded during naturalistic tasks, and has been validated in multiple studies [Huth et al., 2012, Huth et al., 2016, Nishimoto et al., 2011, Cukur et al., 2013, Deniz et al., 2019]. It is the only method that explicitly models the timeseries recorded from the brain, and is thus uniquely suited for analyzing data from continuous tasks. Furthermore, the regression methods are statistically rigorous [Dupre la Tour et al., 2022, Nunez-Elizalde et al., 2019] and are drawn from solid mathematical foundations in linear regression. The comparison with other analysis methods is beyond the scope of this paper and thus we did not include it in the manuscript.
> >
> > Fourth, the reviewer wonders about the credibility, robustness, accuracy, and generalizability of the model. As discussed in the response to point 3 above, VM is a well-established modelling pipeline based on ridge regression, which has solid statistical foundations. To further address these concerns, we have added two new statistical tests for the performance of the model encoding and for the non-random spatial distribution of best-performing LAV modules across the cortex. Please see appendix A.3 for more details about the statistical tests.
> >
> > Finally, concerning the generalizability of the model, we note that the large amount of data we collected from each individual subject is divided into train, validation, and test sets, and models are fit, cross-validated, and tested within each individual subject. Our modelling framework demonstrates that a particular architecture of model, with possibly different parameters per subject that can account for the idiosyncracies of each subject, can be used to accurately explain the data in all subjects. The consistency of our results across subjects is therefore an indication of the generalizability of our model. We have added new figures to appendix A.2 showing the individual subject plots corresponding to the group-level plots in figure 2 and highlighting generalization across different subjects.
> >
> > **References**
> >
> > Asaad, W.F. and Sheth, S.A., 2024. What’s the n? On sample size vs. subject number for brain-behavior neurophysiology and neuromodulation. Neuron.
> >
> > Yamins, D.L., Hong, H., Cadieu, C.F., Solomon, E.A., Seibert, D. and DiCarlo, J.J., 2014. Performance-optimized hierarchical models predict neural responses in higher visual cortex. Proceedings of the national academy of sciences, 111(23), pp.8619-8624.
> >
> > Strong, C., Stocking, K., Li, J., Zhang, T., Gallant, J. and Tomlin, C., 2024, June. A framework for evaluating human driver models using neuroimaging. In 6th Annual Learning for Dynamics & Control Conference (pp. 1565-1578). PMLR.
> >
> > Huth, A.G., Nishimoto, S., Vu, A.T. and Gallant, J.L., 2012. A continuous semantic space describes the representation of thousands of object and action categories across the human brain. Neuron, 76(6), pp.1210-1224.
> >
> > Huth, A.G., De Heer, W.A., Griffiths, T.L., Theunissen, F.E. and Gallant, J.L., 2016. Natural speech reveals the semantic maps that tile human cerebral cortex. Nature, 532(7600), pp.453-458.
> >
> > Nishimoto, S., Vu, A.T., Naselaris, T., Benjamini, Y., Yu, B. and Gallant, J.L., 2011. Reconstructing visual experiences from brain activity evoked by natural movies. Current biology, 21(19), pp.1641-1646.
> >
> > Çukur, T., Nishimoto, S., Huth, A.G. and Gallant, J.L., 2013. Attention during natural vision warps semantic representation across the human brain. Nature neuroscience, 16(6), pp.763-770.
> >
> > Deniz, F., Nunez-Elizalde, A.O., Huth, A.G. and Gallant, J.L., 2019. The representation of semantic information across human cerebral cortex during listening versus reading is invariant to stimulus modality. Journal of Neuroscience, 39(39), pp.7722-7736.
> >
> > Dupre La Tour, T., Eickenberg, M., Nunez-Elizalde, A.O. and Gallant, J.L., 2022. Feature-space selection with banded ridge regression. NeuroImage, 264, p.119728.
> >
> > Nunez-Elizalde, A.O., Huth, A.G. and Gallant, J.L., 2019. Voxelwise encoding models with non-spherical multivariate normal priors. Neuroimage, 197, pp.482-492.

---

> > > ### Comment · Reviewer_csZb · 2024-11-23
> > > **Thanks**
> > >
> > > Thank you for the authors' rebuttal. My main concern remains the lack of technical innovation, and I believe the work does not yet meet the standards of ICLR. Therefore, I will maintain my score.

---

### Official Review · Reviewer_rkWU · 2024-11-03

**Soundness:** 3
**Presentation:** 3
**Contribution:** 3
**Rating:** 5
**Confidence:** 4

**Summary:**

The paper investigates the relationship between human brain activity and deep neural network (DNN) activations during an active driving task, specifically using a simulated taxi-driving environment. By employing functional magnetic resonance imaging (fMRI) to capture brain activity, the authors construct voxelwise encoding models that correlate DNN activations from the Learning from All Vehicles (LAV) model with brain responses. The findings indicate that DNN features can explain significant variance in brain activity across various regions, suggesting a parallel in how both systems process complex sensorimotor tasks. This work represents a new effort to bridge insights from neuroscience and artificial intelligence, particularly in understanding cognitive processes during active driving.

**Strengths:**

The paper's strengths are highlighted by its innovative integration of neuroscience with machine learning, providing valuable insights into how DNNs may emulate human cognitive processes during complex tasks like driving. The rigorous experimental design, which includes detailed comparisons between brain activity and DNN outputs, enhances the reliability of the findings. Additionally, the alignment of DNN modules with specific functional brain regions suggests a meaningful correspondence between artificial and biological systems, indicating potential pathways for future research in both AI development and cognitive neuroscience.

**Weaknesses:**

The findings rely solely on the LAV driving DNN. Testing multiple DNNs trained with different objectives or architectures could strengthen claims about human-AI alignment in driving.

The experiment’s setup, where humans control the stimulus, introduces correlations that may not reflect true alignment in representations, limiting the generalizability of the findings.

While the voxelwise approach is rigorous, the dense presentation and minimal interpretative context might be difficult for a broader ML audience, not sure if ICLR is the best venue for this work.

**Questions:**

Have the authors considered exploring different DNN architectures (e.g., reinforcement learning-based models) to assess if similar regions align across architectures?

Could further studies investigate other interactive tasks, such as social navigation, to see if similar alignment patterns appear in non-driving contexts?

How might the approach handle potential biases from strong correlations in interactive tasks, and are there additional measures to mitigate this?

---

> ### Author Response · Authors · 2024-11-23
> **Response to reviewer rKWU (1/2)**
>
> We would like to thank the reviewer for their helpful comments, which have been valuable for improving the paper. We first provide a brief summary of our changes, then address the reviewer’s points in more detail below. We have performed new baseline comparison experiments in appendix, which we have added to appendix A.2. These experiments show that the LAV DNN features are able to explain more variance in brain activity than those of a standard CNN trained on image classification, especially in high-level vision areas. We also added statistical tests for voxel encoding model performance and for the distribution of best-performing modules across the cortex, which establish the statistical significance of these results. Finally, in order to address questions about the methodology, we have revised section 3 and added appendix A.3 to clarify details about the voxelwise modeling framework as well as why this framework is rigorous and suitable for our analysis.
>
> The reviewer expresses three main concerns. First, they note that we only examined a single driving model, and suggests examining additional models trained with different objectives or architectures. We agree that incorporating comparisons to other driving models to improve our understanding of whether these models converge on similar levels of brain alignment and functional organization is an exciting area. Because running experiments with other models which expect different inputs (e.g. types and positions of cameras) requires rendering different data in addition to repeating regressions for each subject, this isn’t possible for us to address during the rebuttal period, but is an interesting direction for future work. However, we have added new baseline comparisons with CNN models that support the strength of the LAV DNN features at explaining brain activity.
>
> Second, the reviewer expresses concern that correlations between DNN parameters and brain activity do not necessarily imply functional similarity, and rather reflect correlations with other variables. We agree that this is an important consideration. However, correlations between variables are an inherent property of natural environments and naturalistic stimuli. Because both the brain and DNNs learn the statistics of the world, they both will learn these stimulus correlations, and their internal representations will reflect these correlations. It is possible to design the stimulus to control for specific confounds. In vision, for example, one proposed dataset shows subjects images of the same object but with randomly generated backgrounds to control for the effect of the background [Yamins et al., 2014]. However, these types of controls typically reduce the ecological validity of the stimulus, and, because of the nonlinear nature of the brain, may result in brain activity that is not representative of how the brain behaves under more naturalistic conditions. Carefully designing tasks that reduce the influence of specific confounds while maintaining the ecological validity of the stimulus is a promising direction for future work.
>
> Third, the reviewer wonders whether this work is a good fit for ICLR. However, other papers [Benchetrit et al., 2024, Prince et al., 2024] on neuroimaging data have been accepted at ICLR 2024, which also included a workshop on the alignment of representations between artificial systems and biological neural data [Scotti et al., 2024, Nikolaus et al., 2024, Ferrante, et al., 2024].
>
> **Response to questions:**
> 1. Question 1 is about comparisons with other DNN architectures. Please see response point 1 above.
> 2. Question 2 is about whether this approach can be applied to other interactive tasks to better study alignment. To the best of our knowledge, this is the first study that quantifies the alignment between brain activity and a DNN for an interactive task. The methodological framework demonstrated here can be directly applied to other tasks, and this is an exciting direction for future work.
> 3. Question 3 is about possible confounds. Please see response point 2 above.

---

> > ### Author Response · Authors · 2024-11-23
> > **Response to reviewer rKWU (2/2)**
> >
> > **References**
> >
> > Benchetrit, Y., Banville, H. and King, J.R., Brain decoding: toward real-time reconstruction of visual perception. In The Twelfth International Conference on Learning Representations, 2024.
> >
> > Prince, J.S., Fajardo, G., Alvarez, G.A. and Konkle, T., Manipulating dropout reveals an optimal balance of efficiency and robustness in biological and machine visual systems. In The Twelfth International Conference on Learning Representations, 2024.
> >
> > Scotti, P.S., Tripathy, M., Torrico, C., Kneeland, R., Chen, T., Narang, A., Santhirasegaran, C., Xu, J., Naselaris, T., Norman, K.A. and Abraham, T.M., MindEye2: Shared-Subject Models Enable fMRI-To-Image With 1 Hour of Data. In The Twelfth International Conference on Learning Representations Workshop on Representational Alignment, 2024.
> >
> > Nikolaus, M., Mozafari, M., Asher, N., Reddy, L. and VanRullen, R., Modality-Agnostic fMRI Decoding of Vision and Language. In The Twelfth International Conference on Learning Representations Workshop on Representational Alignment, 2024.
> >
> > Ferrante, M., Boccato, T. and Toschi, N., Towards neural foundation models for vision: Aligning eeg, meg and fmri representations to perform decoding, encoding and modality conversion. In The Twelfth International Conference on Learning Representations Workshop on Representational Alignment, 2024.

---

> > > ### Comment · Reviewer_rkWU · 2024-11-28
> > > **response to rebuttal**
> > >
> > > Thank you for the response. However, the reliance on a single DNN architecture and potential confounding variables in the experimental setup significantly limit the robustness of the claims, I'll keep the score the same.

---

### Official Review · Reviewer_cmB2 · 2024-11-03

**Soundness:** 2
**Presentation:** 3
**Contribution:** 3
**Rating:** 6
**Confidence:** 2

**Summary:**

This paper investigates the alignment between human brain activity in the context of autonomous driving and the activations of different modules of a specific deep neural network (Learning from All Vehicles - LAV). Human brain activity was captured in the form of functional magnetic resonance imaging (fMRI), and the alignment was performed through Voxelwise Modeling (VM), previously introduced in the literature. This paper argues that both the deep neural network and the human brain may partition the task of driving in a similar way, by showing that each specific LAV module (semantic segmentation, Bird's-eye-view perception, planning, trajectory prediction, hazard detection, and control) was able to predict different meaningful brain areas.

**Strengths:**

To the best of my knowledge in this applied field, I believe this work surely pushes forward the intersection of neuroscience and machine learning representation; in this sense, and despite the paper "looking different" from typical ICLR papers, I believe this point is in itself a strength of this paper to be accepted at ICLR.

The choice of Learning from All Vehicles (LAV), a competitive model in autonomous driving, strengthens this study’s relevance; LAV’s multi-module structure allowed the authors to link specific network modules to brain regions performing analogous roles. Another significant strength of this work is how this work was devised and how it collected all the data from an actual fMRI machine to be able to explore the active driving paradigm, instead of the more usual passive tasks in previous literature.

In my opinion, this paper is original in its methodological developments and how it tackles a clear gap in the literature with a creative combination of rigorous statistical methods.

**Weaknesses:**

Even though I really enjoyed reading this out-of-the-box paper, and even though I can imagine the insightful discussions this might bring among people attending ICLR, I am afraid this might not be enough for this paper to be accepted at a conference like ICLR. One key point I want to make on this (beyond the weaknesses I list below), is that I believe that a person from the field of neuroscience would be necessary for properly analysing this paper. Section 4 contains a lot of discussions and results focused on brain regions and specific neuroscientific knowledge that I believe it might be difficult to find in ICLR; evaluating this section seems important to me to really understand the contribution and novelty of this paper, which again supports my point that maybe this might not be the best venue for this paper. A more multidisciplinary journal focused on neuroimaging where truly diverse peer reviewers might be easier to find, might be better.

With regards to actual weaknesses that I have found in this paper:
1. In a conference focused on (computational) representation learning, I find that the dataset size of just 3 people is too small for us to trust these results. In order to avoid data leakage, this basically means that one person would be in the training set, another in the validation set for hyperparameter selection, and another in the test size, which in my opinion hinders the potential trust one has in these results as we might not have enough individual variability in brain function in such a complex task like driving. Even though the paper is clearly innovative in its methodological approach, it also contains a clear weakness in providing enough people to truly evaluate its results. Obviously this is not possible to tackle in the rebuttal period, but I think the authors do not provide enough details on how they consider the dataset (small) size in their experiments, and how potential overfitting was avoided.
2. One thing that I believe it's difficult to really evaluate here, and thus it's a weakness of this work, is that these correlations might not necessarily imply functional similarity. Some correlations might come from shared contextual factors (I can think for instance vehicle proximity or visual field overlap) rather than true alignment. I do not know in detail some of the methods applied in this paper, so I was wondering whether the authors could comment on how to potentially tackle this weakness?
3. The paper makes quite a strong statement when it suggests that both the DNN and the human brain may partition tasks in a similar manner. This in itself is a difficult claim to truly evaluate when only looking at one DNN model. I'm not sure whether other autonomous driving models are divided in such well-separated modules, and thus it would be important for the paper to include some discussion on the feasibility of applying this framework into other autonomous driving models currently being used in the real world.

**Questions:**

1. Did the authors use one person for each train/validation/test split used in this paper, to avoid data leakage?
2. How difficult and how long did it take to collect this dataset for these 3 people? How feasible would it be to extend this experiment to a larger number of people to more strongly evaluate this work?
3. Given that the fMRI captures delayed blood-oxygenation responses, do the authors think that a higher temporal resolution imaging method like EEG could help?
4. Isn't the combined $R^2$ of just 0.02 in figure 2 too small to find true alignments between the DNN activations and distinct brain networks? How did the authors choose this value?
5. The paper highlights, in Section 2, some literature connecting fMRI signal with brain activity on driving tasks. Doesn't it mean that the last sentence in introduction ("Our results are an exciting **first step** towards investigating the cognitive and representational basis for human and AI driving") is a bit of an overstatement? (I mean given the usage of the term "first step")
6. In section 3.2.1, the paper mentions some apparent modifications to the original LAV implementation, and that "reasonable inferences" were verified. Can the authors please provide more details on why and how the LAV model was modified, and what "reasonable inferences" mean (eg, how it is defined and evaluated)?

---

> ### Author Response · Authors · 2024-11-23
> **Response to reviewer cmB2 (1/3)**
>
> We would like to thank the reviewer for their helpful comments, which have been valuable for improving the paper. We first provide a brief summary of our changes, then address the reviewer’s points in more detail below. We have performed new baseline comparison experiments in appendix, which we have added to appendix A.2. These experiments show that the LAV DNN features are able to explain more variance in brain activity than those of a standard CNN trained on image classification, especially in high-level vision areas. We also added statistical tests for voxel encoding model performance and for the distribution of best-performing modules across the cortex, which establish the statistical significance of these results. Finally, in order to address questions about the methodology, we have revised section 3 and added appendix A.3 to clarify details about the voxelwise modeling framework as well as why this framework is rigorous and suitable for our analysis.
>
> At a high level, the reviewer expressed concern that a work relating DNN models to neuroimaging data may not be appropriate to the ICLR venue. However, other papers [Benchetrit et al., 2024, Prince et al., 2024] on neuroimaging data have been accepted at ICLR 2024, which also included a workshop on the alignment of representations between artificial systems and biological neural data [Scotti et al., 2024, Nikolaus et al., 2024, Ferrante, et al., 2024].
>
> The reviewer also expressed three more specific concerns:
>
> First, the reviewer is concerned that a pool of three subjects is insufficient for drawing statistically sound conclusions. Here we would like to clarify the n in the conceptual framework underlying our analyses and demonstrate that three subjects is in fact sufficient. The small-n concern expressed by the reviewer reflects the classical psychology experiment framework, in which results from a large number of subjects are averaged to draw a group-level conclusion.
> Neuroimaging experiments under this framework would therefore need to collect data from a large number of subjects, and because of practical limitations, this necessitates collecting less data per subject (typically on the order of one hour per subject). This framework then seeks to create a single model, with particular parameters, for all subjects. However, because of individual differences in anatomy, cognitive strategies, the high-dimensionality of the brain, and the small amount of data collected per subject, these group-level models rarely provide good descriptions of individual subjects and thus the models are of limited use.
>
> Our study instead follows the framework found in psychophysics and neurophysiology (particularly in non-human primates (NHP)) [Asaad et al., 2024]. In this framework, the n is not the number of subjects, but rather the amount of data collected per subject. In our study, we collected 2-3 hours of data per subject in this experiment, and also an additional 5-6 hours of anatomical and functional localizer data that enabled us to reconstruct the cortical surface and delineate known functional regions. This large amount of data from each individual subject is divided into train, validation, and test sets, and models are fit, cross-validated, and tested within each individual subject. Rather than seeking a particular instantiation of a model with particular parameters to apply to all subjects, this framework demonstrates that a particular architecture of model, with possibly different parameters per subject that can account for the idiosyncracies of each subject, can be used to accurately explain the data in all subjects. In other words, the models are fit and statistically tested within each subject, and each subject is in fact a full replication of the experiment [Asaad et al., 2024].
>
> Indeed, studies from NHP neurophysiology and psychophysics under this framework have routinely used as few as two subjects to reveal fundamental insights into the functions of the brain. Neuroimaging studies with small n-in-subjects have also produced robust models of the human brain in complex, naturalistic tasks. Thus, we have in fact provided sufficient data to prevent overfitting and also replicate this experiment. The reviewer commented that we did not consider this in the text of the manuscript, but we note that this n-in-subjects and n-in-data contrast is a philosophical difference between standard practices across fields and is beyond the scope of this paper.

---

> > ### Author Response · Authors · 2024-11-23
> > **Response to reviewer cmB2 (2/3)**
> >
> > Second, the reviewer expresses concerns that correlations between DNN parameters and brain activity do not necessarily imply functional similarity, and rather reflect correlations with other variables. We agree that this is an important consideration. However, correlations between variables are an inherent property of natural environments and naturalistic stimuli. Because both the brain and DNNs learn the statistics of the world, they both will learn these stimulus correlations, and their internal representations will reflect these correlations. It is possible to design the stimulus to control for specific confounds. In vision, for example, one proposed dataset shows subjects images of the same object but with randomly generated backgrounds to control for the effect of the background [Yamins et al., 2014]. However, these types of controls typically reduce the ecological validity of the stimulus, and, because of the nonlinear nature of the brain, may result in brain activity that is not representative of how the brain behaves under more naturalistic conditions. Carefully designing tasks that reduce the influence of specific confounds while maintaining the ecological validity of the stimulus is a promising direction for future work.
> >
> > Third, the reviewer expresses concerns that our claim that the DNN and the brain partition the driving task in a similar manner is difficult to evaluate, given that we have examined only a single DNN in this work. We agree that while it is not straightforward to evaluate the similarity in partitioning, it could nonetheless be evaluated both quantitatively and qualitatively. First, we highlight that encoding models are fit to each voxel independently; the modelling process contains no inductive bias that would encourage models for spatially proximal voxels to have similar partitioning of variance across the different LAV modules. Nevertheless, the best-performing LAV module for each voxel has quantitatively a non-random spatial distribution across the brain. To show that this pattern is statistically significant, we have added a statistical test based on the Moran’s I measure of spatial autocorrelation that finds a p-value of < 0.01 in all three subjects (please see appendix A.3 for more details). This non-random distribution suggests that each model maps to a specific network of functional regions in the brain. Second, the partitioning can be qualitatively evaluated by comparing the LAV module function with the known functional properties of the brain regions to which it is mapped. For example, the semantic segmentation and brake modules, which process RGB images, are the best-performing modules in the visual cortex but are outperformed by planning and control modules in sensorimotor regions. This functional similarity between the LAV module and corresponding brain regions during the same task suggest that they mediate the same aspects of the task. FInally, we agree with the limitation that we have evaluated only a single driving DNN, and that comparison of alignment across DNN architectures will be a key direction for future work.

---

> > > ### Author Response · Authors · 2024-11-23
> > > **Response to reviewer cmB2 (3/3)**
> > >
> > > **Response to questions:**
> > > 1. Questions 1 and 2 refer to the size of n in the experiment, and we refer to response 1 above.
> > >
> > > 3. Question 3 concerns the fact that the BOLD signal recorded by fMRI is delayed by the hemodynamic response, and asks whether higher temporal resolution methods, such as EEG could help. Here, we argue that for both methodological and experimental reasons, other modalities will not provide any benefits. Methodologically, fMRI provides the highest spatial resolution in non-invasive techniques: each voxel directly corresponds to a location in space. Other non-invasive methods, such as EEG, MEG, and fNIRS, all suffer from the source localization problem: each sensor aggregates signal from a large and poorly defined region, and the inverse problem to localize the signal source is ill-defined. Furthermore, these scalp surface-based methods, by the inverse square law, are biased to signal from parts of the cortex that are most proximal to the skull, and thus cannot reliably record signal from medial, temporal, and subcortical regions in the brain. The signal is also attenuated by the skull, hair, and sensor placement. Thus, in studies that seek to relate models to highly localized brain activity, such as ours, fMRI is the optimal imaging method. Experimentally, the process of driving and navigation unfolds over the course of seconds to minutes, and thus is on a timescale commensurate with the fMRI sampling rate. Furthermore, while the BOLD activity is convolved with the hemodynamic response, this delay is accounted for by the finite impulse response filter implemented in the voxelwise modelling process. We do acknowledge, however, that more modern MR pulse sequences with acceleration can increase the sampling rate, and future data collection will make use of better pulse sequences. (Our current data was collected with a water-excite sequence; our fMRI scanner has since been upgraded and can now support multiband sequences with sub-second sampling rates.)
> > >
> > > 4. Question 4 is about the threshold R^2 value in figure 2b. To improve the interpretation for this figure, we replaced the R^2 threshold with a per-voxel statistical significance threshold of p < 0.01. Please see appendix A.3 for more details about the statistical test.
> > >
> > > 5. Question 5 notes that there has been prior work that used fMRI to study the brain activities underlying driving. However, to the best of our knowledge, the prior work on fMRI (and other brain recording modalities) and driving does not include comparisons or connections to AI driving models, and our study is the first to directly compare human brain activity during driving with the activations of an artificial driving system.
> > >
> > > 6. Question 6 is about modifications to the LAV DNN. Thank you for raising this point. In fact, we did not make any modifications to LAV, only to the CARLA simulator that renders the environment (and therefore DNN inputs). We have rewritten this sentence in the updated draft to remove ambiguity.
> > >
> > > References:
> > >
> > > Asaad, W.F. and Sheth, S.A., 2024. What’s the n? On sample size vs. subject number for brain-behavior neurophysiology and neuromodulation. Neuron.
> > >
> > > Benchetrit, Y., Banville, H. and King, J.R., Brain decoding: toward real-time reconstruction of visual perception. In The Twelfth International Conference on Learning Representations, 2024.
> > >
> > > Ferrante, M., Boccato, T. and Toschi, N., Towards neural foundation models for vision: Aligning eeg, meg and fmri representations to perform decoding, encoding and modality conversion. In The Twelfth International Conference on Learning Representations Workshop on Representational Alignment, 2024
> > >
> > > Nikolaus, M., Mozafari, M., Asher, N., Reddy, L. and VanRullen, R., Modality-Agnostic fMRI Decoding of Vision and Language. In The Twelfth International Conference on Learning Representations Workshop on Representational Alignment, 2024.
> > >
> > > Prince, J.S., Fajardo, G., Alvarez, G.A. and Konkle, T., Manipulating dropout reveals an optimal balance of efficiency and robustness in biological and machine visual systems. In The Twelfth International Conference on Learning Representations, 2024.
> > >
> > > Scotti, P.S., Tripathy, M., Torrico, C., Kneeland, R., Chen, T., Narang, A., Santhirasegaran, C., Xu, J., Naselaris, T., Norman, K.A. and Abraham, T.M., MindEye2: Shared-Subject Models Enable fMRI-To-Image With 1 Hour of Data. In The Twelfth International Conference on Learning Representations Workshop on Representational Alignment, 2024.
> > >
> > > Yamins, D.L., Hong, H., Cadieu, C.F., Solomon, E.A., Seibert, D. and DiCarlo, J.J., 2014. Performance-optimized hierarchical models predict neural responses in higher visual cortex. Proceedings of the national academy of sciences, 111(23), pp.8619-8624.

---

> > > > ### Comment · Reviewer_cmB2 · 2024-11-25
> > > >
> > > > I thank the time the authors took to tackle my review, and I want to start acknowledging that I think all my questions are answered.
> > > >
> > > > With regards to weaknesses, I appreciate the authors' comments that other papers which might be similar to this one were accepted at ICLR before, but I hope the authors understand I'm not evaluating this paper based on precedence, but instead on the best I can make out of it. In this sense, I have to admit I still have my reservations on whether ICLR is the best venue for this work because Section 4 of this paper contains a lot of discussions on neuroscientific concepts that I feel are outside the scope of ICLR, both from a reviewing perspective but also for the possible people attending the conference.
> > > >
> > > > Second, I acknowledge the authors comments that the n-in-subjects and n-in-data discussion could be more of a philosophical difference between standard practices across fields. However, regardless of philosophical discussions, the fact is that standard practices in fields can be wrong, and when I say that not considering a different person as a distinct test set is a weakness in this work regarding generalisation, this is not philosophical. For more on this topic in which I'm not just discussing philosophical/standard practices, but actual weaknesses in evaluation that are leading cause of errors in ML applications, I refer the authors to the following study: https://reproducible.cs.princeton.edu/
> > > >
> > > >
> > > > Despite these weaknesses (which I believe were not satisfactorily tackled in this review), I have to admit that any early work on this very complex topic will always be difficult and with their own weaknesses. I do believe the out-of-the-box application of the traditional statistical methods, as well as the ML methods applied, contribute to the originality of this work, and that is why I scored this paper above the acceptance threshold. However, given the other weaknesses mentioned, I'm afraid I cannot increase the score of this work to a clear accept. This is a difficult paper for me to evaluate (a bit outside of my expertise), so I'm also waiting for the remaining reviewers to hopefully still comment on the authors' rebuttal.

---

### Meta-Review · Area_Chair_gpTQ · 2024-12-20

**Metareview:**

The paper studies the relationship between brain activity measured by fMRI during a virtual taxi-driving task with DNN activity. During discussion, the reviewers appreciated that the study opened up an exciting new direction of research. However, they unanimously recommended rejection, citing both insufficient novelty from a technical perspective as well as insufficient completeness and depth of analysis from a neuroscience perspective.

**Additional Comments On Reviewer Discussion:**

This paper generated good engagement and discussion both between the authors and reviewers, as well as between the reviewers and the AC. Post rebuttal period discussion generated a clear consensus summarized in the meta-review.

---

### Decision · Program_Chairs · 2025-01-22

Reject